# GARP and EARP are required for efficient BoHV-1 replication as identified by a genome wide CRISPR knockout screen

**Wenfang S. Tan**[1]*, **Enguang Rong**[1†], **Inga Dry**[1], **Simon G. Lillico**[2,3], **Andy Law**[4], **Paul Digard**[1], **Bruce Whitelaw**[2,3], **Robert G. Dalziel**[1]

**1** Division of Infection and Immunity, University of Edinburgh, Edinburgh, Scotland, United Kingdom, **2** Division of Functional Genetics and Development, University of Edinburgh, Edinburgh, Scotland, United Kingdom, **3** Centre for Tropical Livestock Genetics and Health, the Roslin Institute, Easter Bush Campus, University of Edinburgh, Edinburgh, Scotland, United Kingdom, **4** Division of Genetics and Genomics, University of Edinburgh, Edinburgh, Scotland, United Kingdom

† Deceased.
* wtan2@exseed.ed.ac.uk

**Data Availability Statement:** All data generated in this study have been included in the main files or supplementary files.

## Abstract

The advances in gene editing bring unprecedented opportunities in high throughput functional genomics to animal research. Here we describe a genome wide CRISPR knockout library, btCRISPRko.v1, targeting all protein coding genes in the cattle genome. Using it, we conducted genome wide screens during Bovine Herpes Virus type 1 (BoHV-1) replication and compiled a list of pro-viral and anti-viral candidates. These candidates might influence multiple aspects of BoHV-1 biology such as viral entry, genome replication and transcription, viral protein trafficking and virion maturation in the cytoplasm. Some of the most intriguing examples are VPS51, VPS52 and VPS53 that code for subunits of two membrane tethering complexes, the endosome-associated recycling protein (EARP) complex and the Golgi-associated retrograde protein (GARP) complex. These complexes mediate endosomal recycling and retrograde trafficking to the trans Golgi Network (TGN). Simultaneous loss of both complexes in MDBKs resulted in greatly reduced production of infectious BoHV-1 virions. We also found that viruses released by these deficient cells severely lack VP8, the most abundant tegument protein of BoHV-1 that are crucial for its virulence. In combination with previous reports, our data suggest vital roles GARP and EARP play during viral protein packaging and capsid re-envelopment in the cytoplasm. It also contributes to evidence that both the TGN and the recycling endosomes are recruited in this process, mediated by these complexes. The btCRISPRko.v1 library generated here has been controlled for quality and shown to be effective in host gene discovery. We hope it will facilitate efforts in the study of other pathogens and various aspects of cell biology in cattle.

## Author summary

Bovine Herpes Virus Type I (BoHV-1) causes significant pathology in cows and huge economic losses to cattle farmers worldwide. To aid efforts in controlling BoHV-1 infection,

**Funding:** This study was funded by BBSRC grant BB/P003966/1 to R.G.D. and BBSRC strategic funding to the Roslin Institute via grants BB/P013740/1 and BB/P013759/1 to R.G.D., B.W., A. L. and P.D., as well as an an award to P.D. from the Scottish Funding Council and University of Edinburgh Data-Driven Innovation program. The funders had no role in study design, data collection and analysis, decision to publish, or preparation of the manuscript.

**Competing interests:** The authors have declared that no competing interests exist.

we set out to better understand the interplay between the virus and host cell factors. To this end, we created a large CRISPR knockout (KO) library that is designed to disrupt every protein coding gene in the cow. We then conducted genome wide KO screens during BoHV-1 infection in cells using this library. By doing so we compiled a list of more than 200 genes that, when deactivated by the CRISPRs, lead to less or more efficient replication of the virus. Combining these data with literature searches, we extrapolated candidate complexes or pathways that might facilitate or conversely restrict the virus. We further found that cells deficient in two complexes, the Golgi-associated retrograde protein (GARP) complex and the endosome-associated recycling protein (EARP) complex, released mostly defective viruses that greatly lacked the tegument protein VP8, which is crucial for the virus to establish new infection. Our results suggest important roles these complexes play, along with the membrane vesicles they reside on i.e. recycling endosomes and the previously contended Trans Golgi Network, during cytoplasmic viral protein packaging and envelopment for this virus. The CRISPR KO library generated here was proven effective and should be useful resource in the study of other viruses and many aspects of biology in the cow.

## Introduction

Bovine Herpes Virus Type 1(BoHV-1) infections lead to multiple disease manifestations and major economic loss to cattle industries worldwide [1]. The virus is endemic in the UK, with 80% of herds seropositive for infections [2–4]. Following clearance of acute infection, BoHV-1 establishes lifelong latency in sensory neurons. However, stressful life events can trigger reactivation and clinical diseases. Vaccination can be used to control the disease but strains isolated from respiratory cases and aborted fetuses sometimes correspond to vaccine strains [5]. Improved knowledge of BoHV-1 virology is therefore needed to advance disease control and resistance strategies.

   Like other alpha-herpesviruses, BoHV-1 replicates its dsDNA genome in the host cell nucleus but matures into infectious particles in the cytoplasm. To initiate infection, the viral envelope glycoproteins (glycoprotein B and C) interact with cell surface proteoglycans [6,7] and bring the virus closer to the cellular membrane. This interaction facilitates binding of other viral glycoproteins to host cell receptors [8–11] which subsequently enables cell entry [12]. Once inside the cell, the capsid ventures along microtubules to dock at the nuclear pore, while the tegument proteins execute intricate programs of manipulating host factors in the cytoplasm. After docking, the capsid injects the viral genome into the nucleus to initiate DNA replication and the three stages of viral transcription: immediate early (IE), early (E) and Late (L) [13]. The newly synthesized viral transcripts are shuttled into the cytoplasm for the translation of viral proteins, with capsid subunits transported back into the nucleus for capsid assembly and genome packaging.

   According to the envelopment➞de-envelopment➞re-envelopment model of alpha-herpesvirus replication [14–16], during nuclear egress the capsid assumes a primary envelope at the inner nuclear membrane and is de-enveloped at the outer nuclear membrane. It then buds into a secondary envelope in the cytoplasm via endosomal recycling, gaining the glycoprotein decorated membrane and the tegument protein layer wrapped in between [17]. The events of cytoplasmic packaging and secondary envelopment are complex processes and remain heavily contended topics. Previous studies in human Herpes Simplex Virus Type 1(HSV-1), an alpha-herpesvirus similar to BoHV-1, found that the capsid regains membrane from a cytoplasmic

source decorated with viral glycoproteins, which are first secreted to the plasma membrane [17,18] and retrieved to the cytoplasm via Rab5 GTPase dependent endocytosis [18,19]. They also suggested that membrane tubules of both early endosome [17] and trans-Golgi Network (TGN) [15,20,21] origin can be used to wrap newly synthesized capsids of HSV-1 into infectious virions, although some more recent studies have cast doubt on the TGN being a major source for re-envelopment [17,18]. Comparative ultra-structural studies between HSV-1 and BoHV-1 demonstrated that BoHV-1 assembly utilizes recently endocytosed material for envelopment in a manner analogous to that of HSV-1, while side-by-side analysis further confirmed similar proteome compositions of the BoHV-1 virion to HSV-1 [22]. Although it is known that the post TGN re-envelopment of HSV-1 capsids is Rab5 and Rab11 dependent [17] [19], the detailed mechanisms linking the glycoprotein containing vesicles, such as early endosomes, to the TGN and/or recycling endosomes during secondary envelopment of herpesviruses are poorly understood.

Recent developments in genome editing technologies offer unprecedented opportunities to study host-pathogen interactions. CRISPR/Cas9 in particular has been well purposed for high throughput genetic screens, exemplified by various genome wide CRISPR screens that identified many novel human genes involved in viral infections [23–26], and more recently in pigs [27,28]. Here we present the process of generating a genome wide CRISPR knockout library for cattle, named btCRISPRko.v1, and detail our experience and discovery of almost 300 host factors putatively involved in BoHV-1 infection. We then focused our efforts on validating and investigating some of the top and novel hits. Our results suggest crucial roles for two cellular membrane tethering complexes in tegument protein packaging and secondary envelopment of alpha- and gamma- herpesviruses. The knockout library btCRISPRko.v1 is readily available to study other pathogens in cattle and we think it is a useful resource for the wider cattle research community.

## Results

### A highly transducible MDBK cell line with stable expression of Cas9

To facilitate the screen, we first generated single cell clones with uniform Cas9 expression, using Madin-Darby bovine kidney (MDBK) cells that are highly susceptible to BoHV-1 infection. To achieve this, we designed and produced TAL effector nuclease pair TAL1.6 that specifically target the 1$^{st}$ intron of the bovine rosa26 locus(**Table B in** S1 Text), which is a safe harbour for consistent transgene expression [29]. By co-transfecting MDBKs with the TALENs and a plasmid homology directed repair (HDR) template (**Fig A in** S2 Text), we derived clones that harbour a EF1α promoter driven Cas9 expression cassette in both alleles of the rosa26 following dilutional cloning and PCR genotyping (**Fig A in** S2 Text). Unfortunately, lentiviral transduction efficiency of these MDBK cells was too low to deliver a genome wide CRISPR library cost-effectively with satisfactory coverage (**Fig B in** S2 Text).

An Interferon stimulated gene called TRIM5α has been shown to inhibit RNA viruses including HIV-1 based lentivirus [30–32]. In cows, the TRIM5α locus has undergone multiple rounds of duplication, but within this cluster TRIM5-3 or LOC505265 has been shown to be responsible for the HIV-1 restriction [32]. Thus, we set out to derive bi-allelic knockout clones lacking the B30.2/PRY-SPRY anti-viral domain of TRIM5-3 on top of the Cas9 expression, hoping to increase transduction efficiency in MDBKs (Fig 1A). Due to high sequence similarity between TRIM5-3 and other genes within the TRIM5 cluster [31], we were only able to design three CRISPRs specific to TRIM5-3. We then transfected them into the Cas9+ MDBKs and estimated their editing efficiency by T7 endonuclease I digestion (T7E1). With this assay PCR products amplified from edited cells are digested into two smaller products, in this case

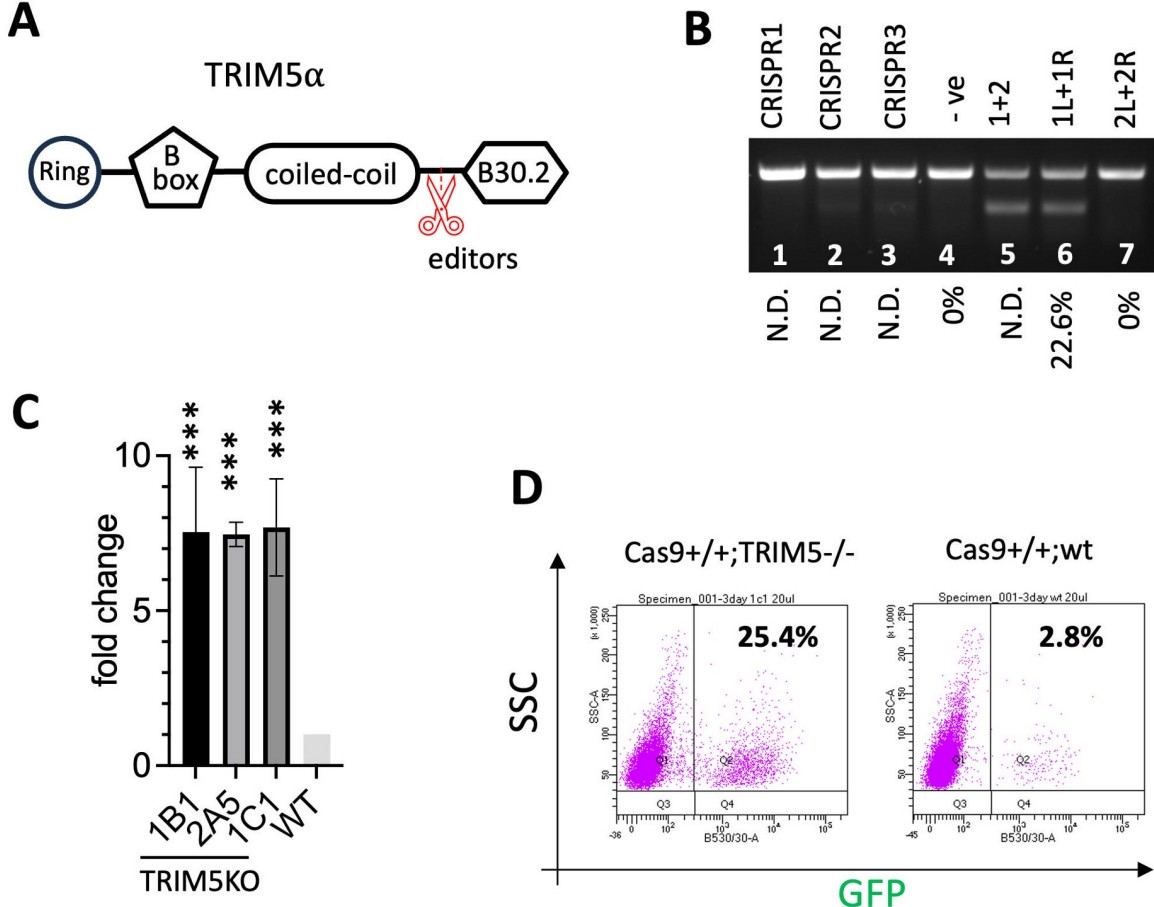

**Fig 1. Production of a highly transducible Cas9 expressing MDBK cell line. A.** Gene editors designed to cut (red scissors) immediately upstream of the B30.2 domain of TRIM5α. **B.** Cutting efficiency of CRISPRs (lanes 1–3) and TALEN pairs (alone in lanes 6–7 or TALEN pair 1 and 2 combined in lane 5) in MDBK cells estimated by T7E1 assay whereby ~200ng of 450bp PCR amplicons of the genomic target from transfected cells were digested by T7 endonuclease I (NEB) and resolved on a 1% agarose gel. Cells transfected with a control plasmid (lane 4, -ve) were used as a control. The efficiency was further quantified by TIDE analysis and shown as percentages at the bottom. N.D.: not determined. **C, D.** Three clones homozygous for Cas9 KI and TRIM5α KO (Cas9+/+; TRIM5-/-) transduced with GFP lentivirus and the transduction efficiency measured by FACS (n = 3; ***: p<0.001). Fold changes in percentage of GFP+ cells from transduced Cas9+/+; TRIM5-/- cells (clones 1B1,2A5,1C1) relative to WT cells were shown in bar chart (**C**). Representative FACS plots from clone 1c1 and Clone P (WT control) (**D**).

presented as two overlapping bands on the agarose gel. Compared to a single larger product amplified from unedited cells, the intensity of digested PCR products correlates to the percentage of edited cells. Unfortunately, based on our T7E1 results, none of these CRISPRs demonstrated good cutting efficiency at this locus (Fig 1B). We then designed and synthesized two pair of TALENs, and after transfection one pair (1L+1R, **Table B in** S1 Text) lead to efficient and specific targeting of TRIM5-3, also assayed by T7E1 (Fig 1B). We were then able to isolate Cas9+/+;TRIM5α -/- single cell clones with homozygous Cas9 expression and TRIM5-3 deletion (**Table E in** S2 Text). As a result, the transduction efficiency in these clones increased eight-fold (Fig 1C and 1D) while its BoHV-1 infectibility was fully maintained compared to WT cells (**Figs C-E in** S2 Text), indicating that Cas9 over-expression and TRIM5α knockout do not interfere with BoHV-1 replication. We thus conducted our BoHV-1 screens in these Cas9+/+;TRIM5α -/- MDBK cells and validated the candidates in WT cells or a Cas9+/+ only clone (Clone P in lane 3, **Fig A in** S2 Text).

## CRISPR library designed with optimal on-target efficiency and limited off-targeting *in silico*

To make a genome wide knockout library broadly applicable to high throughput functional genomics research in cattle, we complemented the UMD3.1.1 genome assembly with the Y chromosome from the btau5.0.1 genome assembly. For each coding gene, sequences shared among all RefSeq transcript isoforms were scanned for 20bp sequences followed by NGG, the Cas9 protospacer adjacent motif (PAM) (Fig 2A). We then filtered all CRISPRs using our initial design criteria and only kept those that: a. had a 20–80% GC content; b. did not harbour BbsI binding sites "GAAGAC" and "GTCTTC", and c. did not contain any of the following sequences "N", "AAAA", "TTTT", "GGGG", and "CCCC" (S1 Text). Each remaining CRISPR was then estimated for the on-target efficiency using Azimuth 2.0 [33] and aligned back to the genome to identify all potential off-targets using BWA [34]. For each gene, 4–5 top candidate guides ranked by on-target efficiency, cut position and off-target specificity (S1 Text) were picked into a final library totalling 94,000 guides targeting 21,165 genes, along with 2000 non-

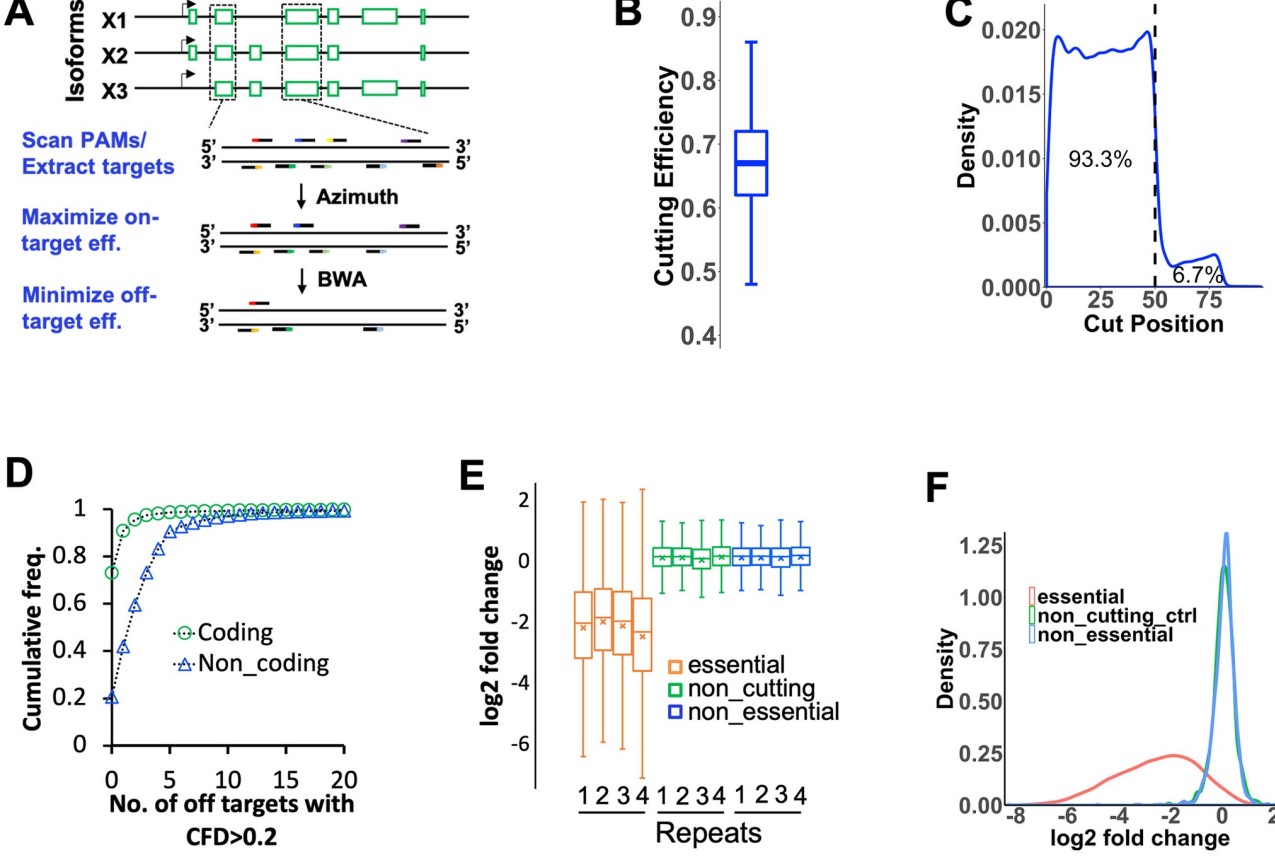

**Fig 2. Library design, statistics and performance assurance. A.** pipeline developed for the genome-wide CRISPRko library design; **B.** Distribution of cutting efficiency for all guides estimated by Azimuth 2.0 (S1 Data); **C.** Frequency of the cut positions, calculated as percentages of the full length peptides relative to the 5' of common coding sequences of targeted genes for all guides (S1 Data); **D.** Cumulative frequency of guides that have zero to 20 off-targets with CFD scores > = 0.2 within coding or non-coding sequences (S1 Data); **E.** Distribution of log2 fold changes (l2fc) in copy numbers of guides targeting CEG2.0 genes(orange), non-essential genes(blue) and non-targeting gRNA controls(green) in cells transduced with the lentiviral K2g5 library compared to the plasmid library (n = 4, S3 Data). For each repeat, the boxes span from the 1st Quartile (Q1) to the 3rd Quartile (Q3), with the median represented by a line and mean by a cross; the lengths of the upper and lower whiskers are equal to 1.5*(Q3-Q1). **F.** Distribution of averageed l2fc of all guides in each group (n = 4, S3 Data). The data was plotted using a modified python script based on what's published by Hart et al. 2017 [44].

targeting controls (S1 Data **and Table A in** S2 Text). Taken together these guides have a median cutting efficiency of 0.67 based on calculations by Azimuth 2.0 (Fig 2B) and 93.3% of all targeting guides cut in the first half (0–50% cut position) of common coding sequences (Fig 2C). Also 73% of all guides have zero predicted off-targets residing in coding regions with a CFD score > 0.2 (Fig 2D) [33] and less than three such off-targets in non-coding regions (Fig 2D).

## Library production and performance assurance in MDBK cells

Prior to library synthesis, we validated the design by testing guides against seven host genes [35–38] and all guides induced indels at the intended sites, with cutting efficiency up to 79% (**Fig F in** S2 Text). We then synthesized the library as one ssODN pool and cloned it into two almost identical lentivirus plasmids [39,40] containing either the original sgRNA scaffold from the Broad Institute (**Fig G and Table B in** S2 Text) [41] (referred as the K2g2 library) or an optimized version from Chen et. al [42] (referred as K2g5). After packaging and trans-ducing the two libraries into Cas9+/+;TRIM5α -/- MDBKs with Puromycin selection, we PCR amplified the CRISPR regions using genomic DNA from the cells as templates and sequenced the amplicons by NGS (S1 Text **and Figs H and I in** S2 Text). After data analysis using purposely built CRISPR screen analysis package MAGeCK [43], we saw a higher deple-tion rate of guides targeting core essential genes (CEG2.0) [44] in K2g5 transduced cells than those by K2g2 (**Fig J in** S2 Text **and** S2 Data), pointing to superior performance of K2g5. Re-sequencing of the K2g5 cells with higher sequencing depth confirmed the shift (Fig 2E and 2F **and** S3 Data) and the relative unaltered representation of non-cutting controls, suggesting good off-targeting control during the library design. We also detected 99.9% of all guides in the plasmid library with a GiniIndex of 0.1258 [43], and 96.01% of all sequencing reads mapped to the library without any mismatch (S3 Data). These data reflect good coverage and distribution, and accuracy of the K2g5 library. Thus, we chose the K2g5 library for our BoHV-1 screens. To expand the usefulness of the library, we also cloned these guides into a piggyBac vector to enable delivery of the library into hard to transduce cells (**Fig K in** S2 Text).

## Overview of the genome wide CRISPR knockout screens

We first obtained a GFP tagged BoHV-1 virus strain [45] and developed a reliable FACS assay capable of isolating cells with differential levels of BoHV-1 replication based on GFP (Fig 3 **and Fig L in** S2 Text). This strain was derived from BoHV-1.1(strain Jura), the most prevalent respiratory form around the world [46], with the GFP fused to the C terminus of VP26 [45]. A full cycle of BoHV-1 infection takes 8–15 hours to complete; VP26 is a minor capsid protein synthesized during the L stage of transcription. After infecting library trans-duced cells with this virus, we FACS collected four fractions of live cells with Negative (Neg), Low, Medium, and High levels of GFP intensity (Fig 3). We then PCR amplified and sequenced the CRISPR regions as before and obtained the copy number of all CRISPRs by counting matching reads. Using MAGeCK [43], we pair-wise compared CRISPR copy numbers between fractions, and identified genes with significantly enriched or depleted guides, to which anti- or pro-viral roles were assigned. We then conducted a focused CRISPR knockout screen of the top candidates for quick validation, followed by pathway analyses and literature search to uncover the most interesting candidates for further investigation.

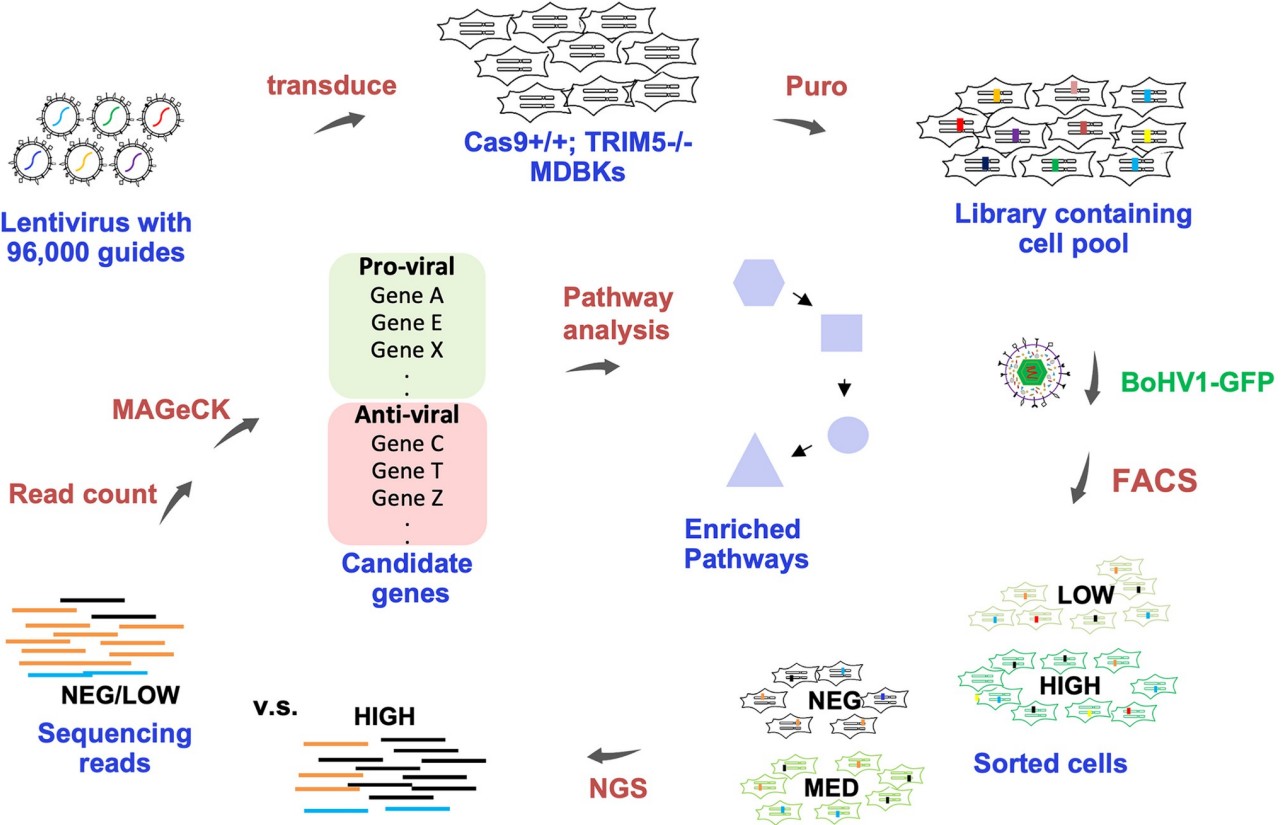

**Fig 3. Overview of the CRISPR KO screen.** Genome wide CRISPR KO library is packaged into lentivirus and transduced into Ca9+/+; TRIM5α-/-MDBKs. After Puromycin selection, the transduced cells are challenged with a GFP tagged BoHV-1 virus and FACS sorted into four groups based on GFP intensity, GFP Negative, Low, Medium, and High. Genome DNA is isolated from recovered cells and used as template for PCR to amplify the CRISPR regions. The PCR products are sequenced by Illumina NextSeq and trimmed reads matching each guide in the library are counted. With MAGeCK, read counts from each intensity group are pair-wise compared to identify enriched or depleted guides and their target genes/pathways.

## CRISPR KO screen identifies pro- and anti-viral candidates involved in many aspects of virus biology

In total we conducted two rounds of genome wide screens. Initially we infected cells with an MOI = 2 and sorted them at 10 hours post infection (10 h.p.i) (**Fig M and Table C in** S2 Text). The experiment was repeated three times and after sequencing and data analyses, we identified genes with well-established roles in BoHV-1 replication as well as many undocumented candidates (S3 and S5 Data files). Encouraged by the results, we performed a 2nd screen by FACS sorting cells harvested at 8 h.p.i. The earlier time point allowed a more stringent gating strategy (**Fig M in** S2 Text), capturing more extreme phenotypes but also higher numbers of cells returned by the FACS sort as the infection was less advanced and cells were healthier (**Table D in** S2 Text). Reassuringly, this screen recapitulated many candidates highlighted in the 1st screen, along with new candidates (Fig 4A and S4 and S5 Data files). There was a good regression trend between the l2fc of common pro-viral candidates from the two screens (Fig 4B and S3, S4 and S5 Data **files**), indicating screen reproducibility.

For both screens, comparisons of either GFP Neg or Low cells with the High cells lead to the most candidates. With strict cut-offs (adjusted p-value < = 0.005, FDR< = 0.1 and log2 Fold Changes(l2fc) > = 0.95), a total of 154 pro-viral and 138 anti-viral candidate genes

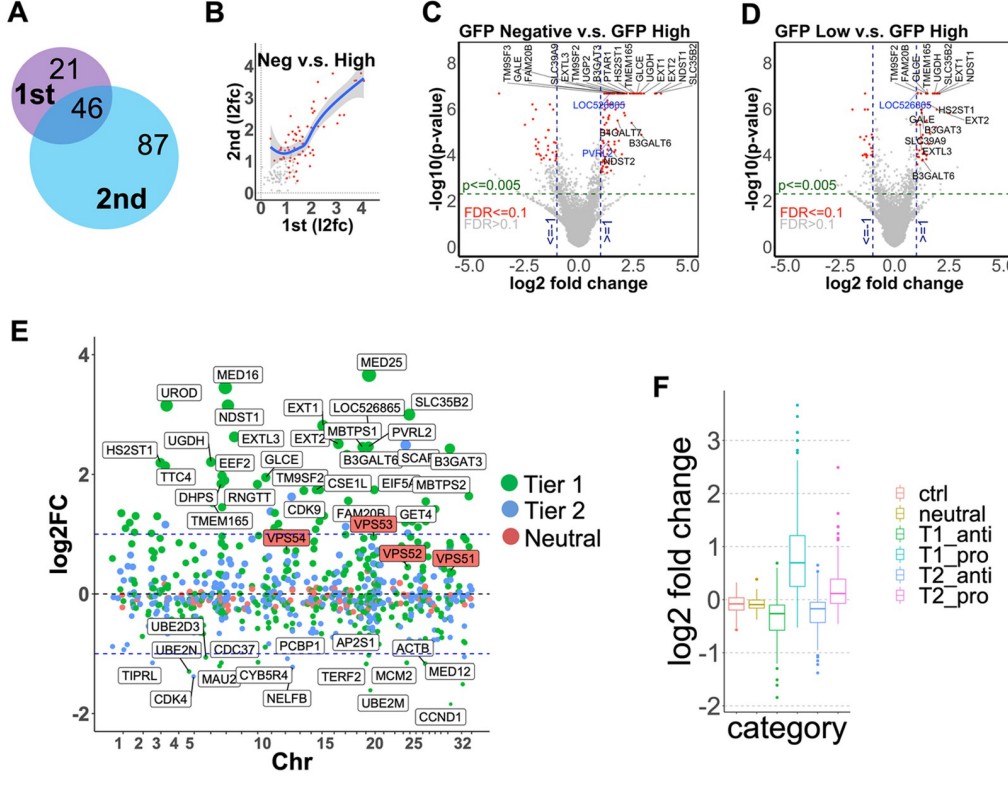

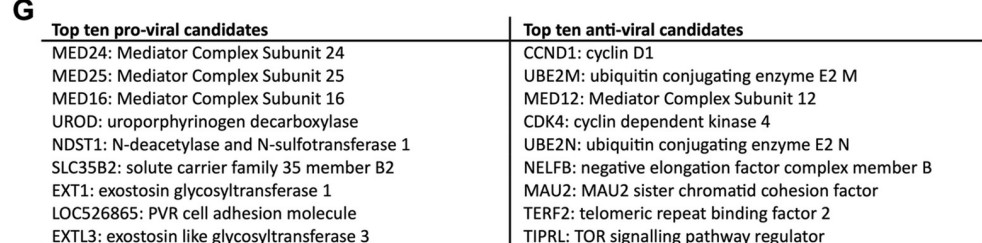

**Top ten pro-viral candidates**

| Top ten pro-viral candidates | Top ten anti-viral candidates |
|---|---|
| MED24: Mediator Complex Subunit 24 | CCND1: cyclin D1 |
| MED25: Mediator Complex Subunit 25 | UBE2M: ubiquitin conjugating enzyme E2 M |
| MED16: Mediator Complex Subunit 16 | MED12: Mediator Complex Subunit 12 |
| UROD: uroporphyrinogen decarboxylase | CDK4: cyclin dependent kinase 4 |
| NDST1: N-deacetylase and N-sulfotransferase 1 | UBE2N: ubiquitin conjugating enzyme E2 N |
| SLC35B2: solute carrier family 35 member B2 | NELFB: negative elongation factor complex member B |
| EXT1: exostosin glycosyltransferase 1 | MAU2: MAU2 sister chromatid cohesion factor |
| LOC526865: PVR cell adhesion molecule | TERF2: telomeric repeat binding factor 2 |
| EXTL3: exostosin like glycosyltransferase 3 | TIPRL: TOR signalling pathway regulator |

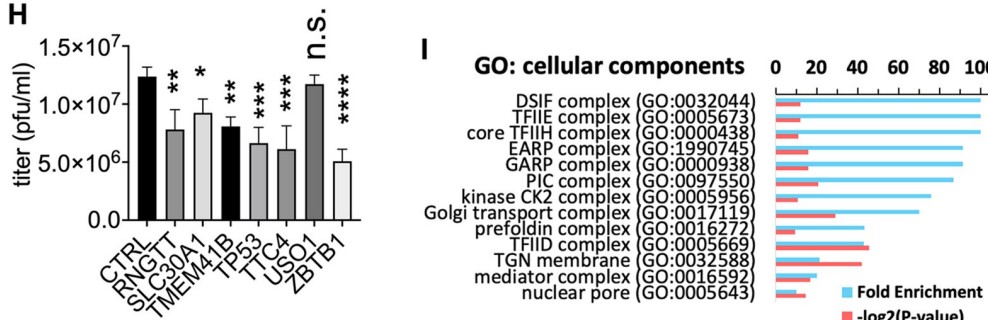

**Fig 4. Candidate host genes identified by the CRISPRko screens. A.** Venn diagram showing numbers of unique and common pro-viral candidates identified by the 1st and 2nd screens (S5 Data). **B.** l2fc of guide copy numbers for all common pro-viral candidate genes (pos.p-value < = 0.05 based on the MAGeCK results comparing GFP Neg. cells to High cells) from the 1st and 2nd screens plotted (S3 and S4 Data **files**). Candidates with pos.p-value< = 0.005, FDR <0.1 and l2fc> = 1 for both screens are highlighted in red. The trendline was drawn using Local Polynomial Regression Fitting with 0.95 as level of confidence interval. **C,D.** Volcano plots of l2fc of guide copy numbers for all genes in the GFP Neg (**C**) or Low (**D**) cells compared

to High cells from the 2nd screen. Candidates with p<0.005, FDR<0.1 and abs(l2fc)> = 1 are highlighted in red (S4 Data). Candidate entry factors are labelled in blue (surface receptors) or black (genes involved in HS biosynthesis). E,F,G. Focused screen results with genomic coordinates of all of genes targeted (tier 1, tier 2 and neutral genes) on the x-axis and their l2fc of guide copy numbers in the GFP Neg. v.s. High cells on the y-axis (n = 3). Some of the top candidates or the GARP/EARP subunits (red labels) were highlighted(E). Results plotted as boxplot with tier 1 and 2 genes divided into pro- or anti-viral candidates. Each box represents the 1st Quartile (Q1) to the 3rd Quartile (Q3) of l2fc for all genes within that category, with the median marked by a line. The whisker lengths are equal to 1.5*(Q3-Q1) and the dots represent outliers(F). The top ten genes that are not on CEG2.0 with p<0.005 and FDR< = 0.1 and the highest (pro-viral) or lowest (anti-viral) l2fc values are listed (G). H. CRISPRi validation of candidates with virus titers obtained by direct plaquing. Significant difference between CRISPRi cells and controls were highlighted (****: p<0.0001; ***:0.0001 to 0.001; **:0.001 to 0.01; *: 0.01 to 0.05). I. Significantly enrich cellular components by Gene Ontology analysis shown with -log(p-value)> 10 and Fold Enrichment > 10 (S7 Data).

emerged (S3, S4 and S5 Data **files**, highlighted in blue in S5 Data). Guides targeting pro-viral genes were enriched up to 25-fold in the GFP Neg and Low cells, (l2fc = 4.66), and those targeting anti-viral candidates were enriched up to 13-fold in GFP High cells (l2fc = 3.65). The 2nd screen re-identified 46 out of the 67 pro-viral candidates from the 1st screen, together with 87 new candidates (Fig 4A and S5 Data). In total, the list combines 50 pro-viral and 45 antiviral candidates identified from the 1st screen (S3 and S5 Data **files**), as well as 116 pro-viral and 53 anti-viral genes from the 2nd screen (Fig 4C and 4D, highlighted in red and S4 and S5 Data **files**).

## Validation of candidates by a focused CRISPR KO screen and by CRISPRi

We then conducted a focused screen for quick validation and re-ranking of the top candidates. Candidate genes selected with p-value < 0.05 and l2fc > = 0.75 (pro-viral) or l2fc < = -0.75 (anti-viral) from the GFP Neg/Low v.s. High groups (based on the 2nd screen results) were further divided into two tiers, with tier 1 genes satisfying more stringent statistics (p-value< 0.005 and l2fc > = 0.95 or l2fc< = -0.95, S1 Text). We then produced a mini-KO library containing 6,071 guides targeting 679 candidate genes and 83 neutral genes that ranked between 10,000th and 13,000th places by MAGeCK with l2fc falling in the range of (-0.06,0.06), with eight guides per gene, as well as 326 non-targeting controls (S6 Data). After library packaging, transduction and Puro selection, cells containing the focused library were infected and sorted using the same BoHV-1 challenge and FACS gating strategy as the 2nd genome-wide screen. Upon comparative analyses of the read counts of guides in GFP Neg and High cells, we observed expected CRISPR distribution patterns for most candidates with positive l2fc values for pro-viral candidates and negative l2fc for anti-viral genes in GFP Neg cells (Fig 4E and 4F). Also, CRISPRs targeting tier 1 candidates had a more extreme distribution of l2fc of CRISPR copy numbers than those targeting tier 2 candidates, e.g., 64.6% of guides targeting tier 1 pro-viral candidates (T1_pro) and 25.2% for tier 2 pro-viral candidates (T2_pro) fell outside the range of neutral genes (Fig 4F **and** S6 Data). These data validate the candidacy for many genes tested, with the top ten validated pro- and anti-viral genes listed in Fig 4G, providing further evidence that they play a role in BoHV-1 replication.

We also sought to test some of the candidates using an orthogonal method called CRISPR interference (CRISPRi). CRISPRi uses a catalytically deactivated Cas9(dCas9) fused to transcriptional interfering domains, such as the KRAB and the TRD domain of MeCP2 [47]. When the dCas9 is coupled with sgRNAs that bind immediately downstream of transcription start sites (TSSs), the dCas9/sgRNA complex can block procession of the gene transcription machinery, achieving gene knockdown [47]. To validate genes using CRISPRi, we first random selected seven candidate genes, RNGTT, SLC30A1, TMEME41B, TP53, TTC4, USO1 and ZBTB1. For each gene, we designed three CRISPR sgRNAs that target the TSS of its principal transcript (**Table C in** S1 Text). We then expressed these three sgRNAs simultaneously in

MDBK cells with Doxycycline inducible dCas9-KRAB-MeCP2 expression as previously described [47] and performed plaque assays on them directly. We found that multiplexed CRISPRi knockdown of all genes targeted, except for USO1, significantly reduced efficiency of virus plaque formation compared to control cells transfected with three non-targeting guides (Fig 4H).

## Gene ontology analyses for enriched pro-viral cell components and biological processes

To prioritize further validation efforts, we conducted gene ontology (GO) analysis on the pro-viral candidates. With Fold Enrichment >10 and -log2(P-value) >10 as cut-offs, 13 cellular components (Fig 4I) and 30 biological processes (**Fig N in** S2 Text) stood out (S5 and S7 Data files). Both lists reflect diverse biological functions occurring at different locations in the host cell. Some of these candidates are obvious and already highlighted by literature [10,48], such as processes at different steps of Heparan Sulfate (HS) synthesis (e.g., GO:0006065, GO:0015014, and GO:0030210) and general transcription factors indispensable for viral transcription (e.g. GO:0005673, GO:0000438, GO:0051123, GO:0006368), while others require substantial work to pinpoint their possible function in BoHV-1 biology. Two components captured our attention further, the endosome-associated recycling protein (EARP) and the Golgi-associated retrograde protein (GARP) complexes (GO:1990745 and GO:0000938), being the 4[th] and 5[th] most enriched cellular components according to GO (Fig 4I). Both of them have been shown to be important for endocytic recycling [49] but no roles in alpha-herpesvirus replication have been described.

## Anti-viral candidates identified by CRISPR KO screen

Initial analysis identified 56 anti-viral candidate genes from the screens and literature search highlighted several candidates of interest. For example, FBXW11 and TFDP1 that regulate cell cycle progression, and CDC45, MCM2 and UBE2M for NEDD8-conjugating enzyme UBC12 (S5 Data). Re-analysis of the primary screens by slightly relaxing the statistics (p< = 0.05, FDR< = 0.2 and l2fc < = -0.9) expanded the families of genes that regulate cell cycle and DNA replication and highlighted other sets of genes with good anti-viral potential (86 more candidates with an average l2fc = -1.28, S5 Data). In particular, a group of genes associated with the HOPS/CORVETT complexes that coordinate late endosome and lysosome fusion, i.e. VIPAS39, VPS16, VPS18, and VPS41, all became significant (Fig 5). In addition, the screen pinpointed other E2 ubiquitin-conjugating enzymes or ubiquitin related or like proteins, including UBA52, UBL5, UBE2D3, with the latter shown to be essential for RIG-I and MAVS aggregation in antiviral innate immunity [50]. Two negative regulators of mTORC1 signalling, TSC1 and TSC2 that are phosphorylated by Us3 during HSV-1 infection resulting in mTORC1 activation [51], [52], were also identified.

## Genes with potential roles in BoHV-1 cytoplasmic and nuclear entry

We recovered PVR (LOC526865 or nectin-1) and PVRL2 (nectin-2, Fig 4C and 4D, blue labels, S5 Data), two membrane proteins previously implicated as receptors for BoHV-1 [48,53,54]. The screens also captured at least 20 genes involved in crucial steps of Heparan Sulfation [55] (Fig 4C and 4D, black labels, **Figs O and P in** S2 Text). Another group of candidates, COG1, COG2, COG4, COG5, COG6 and COG7, codes for six of the eight subunits of the Conserved Oligomeric Golgi Complex (COG), an evolutionarily conserved peripheral membrane protein complex residing within the Golgi apparatus. It is thought to act as a

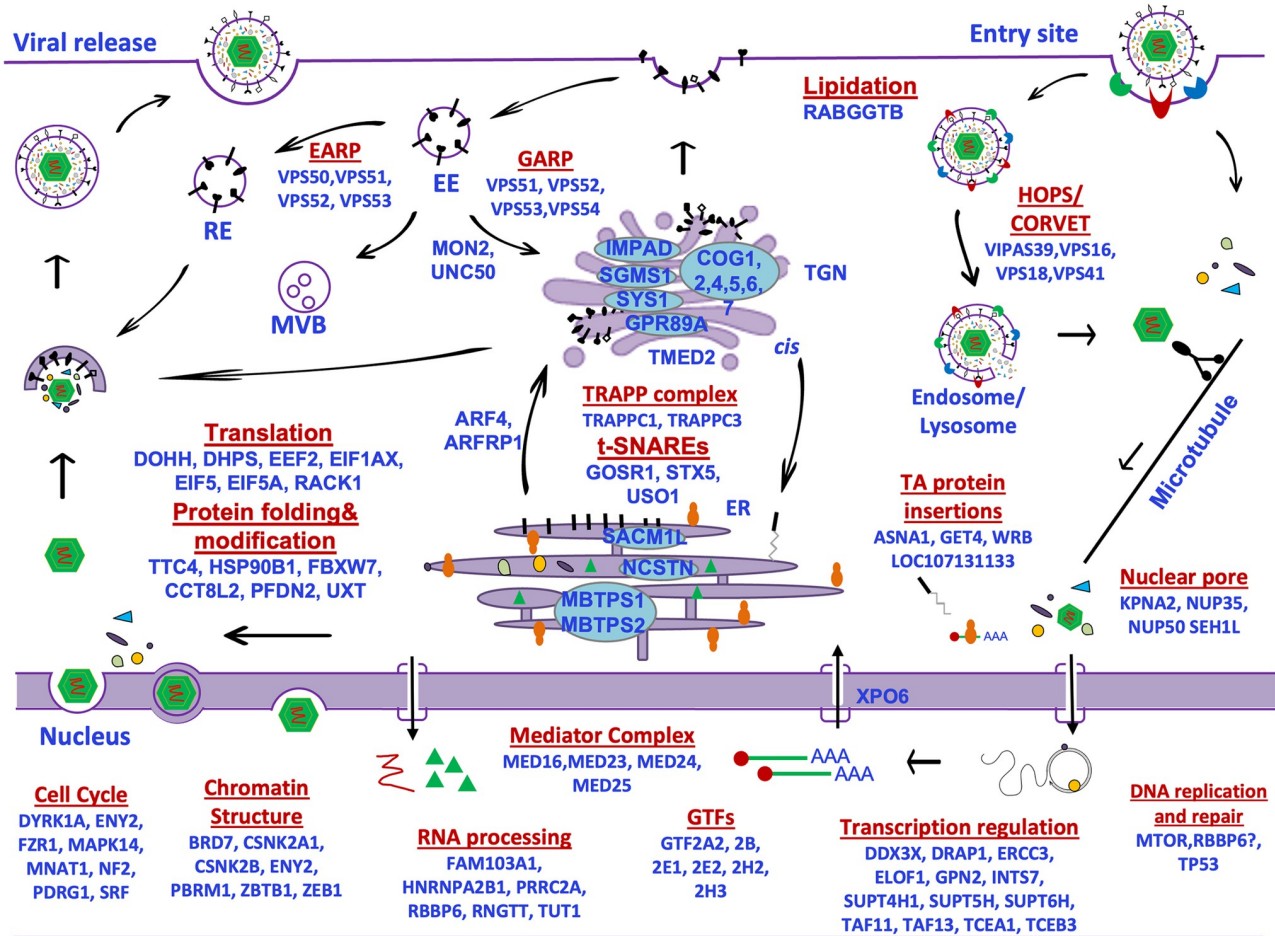

**Fig 5. Diagram highlighting candidate pro- and anti-viral host genes involved in multiple steps of the virus replication cycle.** Diagram depicts the complete life cycle of the virus from entry to cytoplasmic egress with candidates indentified by the screens (p <0.005, FDR < = 0.1 and |l2fc| > = 1 as selection criteria) highlighted at each step based on literature search. Due to space constraints, not all candidates identified are shown (for a full list of candidates, please refer to S5 Data).

retrograde vesicle tethering factor in intra-Golgi trafficking [56] and be crucial for glycoprotein modification and HS secretion to the cell surface [57]. The identification of many such proteins known to be important to BoHV-1 and multiple subunits from the same complexes and pathways demonstrate robustness of the screens.

The BoHV-1 capsid docks at the nuclear pore complex (NPC) for genome delivery after cell entry. As the gateway for bi-directional and selective transport of nucleic acids and peptides, the NPC consists of 34 distinct nucleoporins embedded in the nuclear membrane, of which NUP35, NUP50 and SEHL1 were identified by the screen (Fig 5 **and** S5 Data). Another hit, KPNA2 encoding importin α1, has been shown to mediate import of several HSV-1 proteins into the nucleus and modulate assembly and egress of its capsids [58]. XPO6, a gene encoding Exportin-6 that acquires cargo in the nucleus and shuttles it to the cytoplasm via the NPC, was also discovered. How these proteins participate in the bi-directional transport of BoHV-1 transcripts and proteins between the cytoplasm and the nucleus remains unknown.

## Genes related to genome replication, gene transcription, RNA processing and translation also identified

Alpha-herpesviruses have evolved to usurp host transcription machinery to express viral genes and manipulate host transcription. At least eight factors within the RNA polymerase II transcription pre-initiation complex (PIC) were significantly depleted in the GFP Neg and Low cells (Fig 5 and S5 Data). In addition, genes encoding subunits of the Mediator complex proven to interact with ICP4 of HSV-1 [59], namely MED16, MED23, MED24 and MED25, were among some of the most depleted (Figs 4G and 5 and S5 Data). Genes known to regulate RNA elongation and processing post transcription initiation, e.g. DDX3X [60], DRAP1, ERCC3, GPN2, INTS7, SUPT4H1, SUPT5H, SUPT6H [59], TCEA1 and TCEB2, were also significantly deleted. Furthermore, multiple genes involved in chromatin restructuring, namely BRD7 [61], CSNK2A1, CSNK2B, ENY2, PBRM1 [61], ZBTB1, and ZEB1, were also identified. Many of these gene products have been found to influence HSV-1 replication and transcription [59]; these results indicate BoHV-1 may co-opt these proteins in a similar fashion to HSV-1.

During and post transcription, precursor mRNA (pre-mRNA) transcripts are capped, poly-Adenylated, and spliced prior to nuclear export. Many animal viruses appropriate host factors to achieve efficient viral translation or divert host RNA for degradation via virion host shutoff (vhs) [62]. Indeed, several genes including FAM103A1 and RNGTT important for mRNA capping [63], HNRPA2B1 [64] and PRRC2A [65] for m6A-dependent nuclear RNA processing, and RBBP6 for pre-mRNA processing were all preferentially deleted in GFP Neg and Low cells (Fig 5 and S5 Data). TUT1, a gene encoding a protein that directly adds 3' polyA tails to mRNA and polyU tails to miRNAs and snRNAs, the latter being important for pre-mRNA splicing [66], was also depleted.

Once transcribed and modified, most viral transcripts are translated into peptides that are folded prior to viral packaging and egress. Protein translation is a multi-step process consisting of initiation, chain elongation and termination. Our screen identified multiple factors involved in these processes, namely EEF2, EIF1AX, EIF5, EIF5A. Interestingly, genes encoding two proteins catalysing the conversion of lysine to the unique amino acid hypusine in EIF5A [67], DOHH and DHPS (Fig 5 and S5 Data), were significantly depleted in GFP Neg and Low cells from both screens. These findings potentially highlight the prime importance of EIF5A in modulating BoHV-1 protein translation. In addition to translation initiation and control, genes important for proper folding of newly synthesized peptides, e.g. TTC4, HSP90B1, PFDN2, were also preferentially deleted in GFP Neg and Low cells.

## Single gene VPS51, VPS52, and VPS53 knockout severely impairs production of infectious particles

Three sets of candidates highlighted by STRING protein analysis promote intracellular protein and membrane trafficking [68]. One cluster consists of VPS51, VPS52, and VPS53 (Fig 6A, red nodes) that form part of GARP, a retromer complex that tethers retrograde endosomal carriers to the TGN [49]. Genes known to regulate this retrograde trafficking process, namely SYS1, ARFRP1, and RGP1, were also enriched. The fourth subunit of GARP, VPS54 primarily locates the complex to the TGN and to recycling endosomes (REs) to a much lesser extent [49]. Replacing VPS54 with VPS50 forms another complex, EARP that mainly resides on REs [49]. The other cluster code for tethering or regulatory factors essential for intra-Golgi transport or membrane trafficking between the Endoplasmic Reticulum (ER) and the Golgi Apparatus (Fig 6A, green nodes). These include members of tethering SNAP Receptors or t-SNAREs (GOSR1

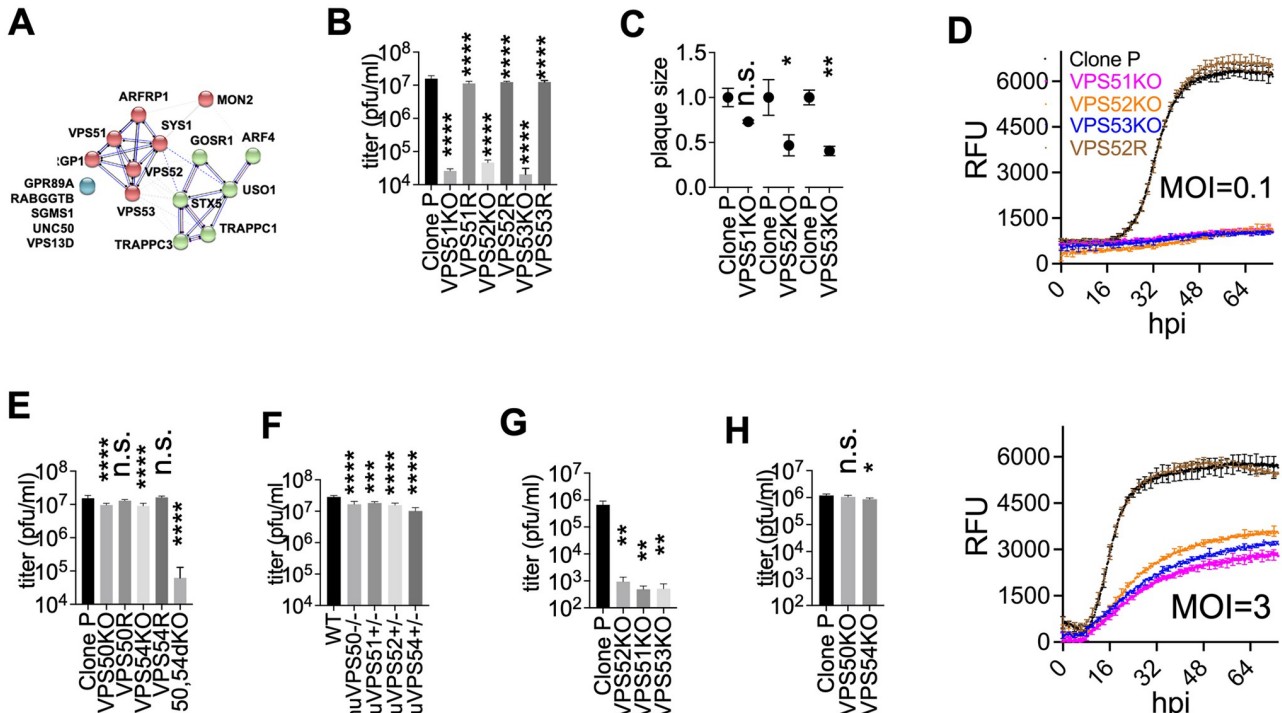

**Fig 6. Dual loss of GARP and EARP leads to impaired virion production and these two complexes play alternative roles. A.** STRING analysis and clustering of candidates potentially involved in intra-cellular trafficking. Blue dots represent disconnected genes based on the current STRING database 11.0. **B,C.** Plaque formation by direct plaquing of BoHV-1 on VPS51KO, VPS52KO and VPS53KO clones and their corresponding rescue clones (VPS51R, VPS52R and VPS53R) with calculated titers (**B**) and plaque sizes (**C**) shown (n = 4). **D.** VP26-GFP growth tracked over 72h in control cells (Clone P), VPS51KO, VPS52KO, VPS53KO and rescue cells (52R) infected with BoHV-1 at MOI = 0.1 or MOI = 3. **E.** Direct plaquing of BoHV-1 on VPS50KO, VPS54KO and 50,54dKO (VPS50 and VPS54 double KO) cells with titers shown. **F.** HSV-1 titration in various A549 clones deficient in GARP and EARP. **G,H.** Direct plaquing of AIHV-1 on VPS51KO, VPS52KO, VPS53KO (**G**) or VPS50KO and VPS54KO MDBK clones (**H**) with calculated titers shown. Significant differences between genotypes are highlighted (****: p<0.0001; ***:0.0001 to 0.001; **:0.001 to 0.01; *: 0.01 to 0.05).

and STX5), regulator of SNARE assembly required for ER-to-Golgi transport (USO1 and ARF4) and subunits of the transport protein particle tethering complex or the TRAPP complex (TRAPPC1 and TRAPPC3). There are also nodes disconnected from the network with suspected roles in intra-cellular trafficking, namely GPR89A, RABGGTB, SGMS1, UNC50, and VPS13D (Fig 6A, blue nodes, S5 Data).

To probe the importance of the GARP/EARP in BoHV-1 replication, we made a series of KO cells missing one of the subunits VPS51, VPS52 or VPS53. By plaquing BoHV-1 directly on these KO clones (**Table E in** S2 Text), the apparent virus titre dropped by almost 3-log compared to non-KO control (Clone P) cells (Fig 6B). The plaques also became smaller, with up to 59% reduction in size compared to those grown in the control cells (Clone P, Fig 6C). Overexpressing cDNA encoding the missing subunits in the KO cells (VPS51R, VPS52R, VPS53R) greatly rescued the impairment in both viral titre and plaque size (Fig 6B **and Fig Q in** S2 Text). We also tracked the rate of VP26-GFP synthesis in these KO cells by measuring GFP fluorescent intensity from 0 h.p.i. to 72 h.p.i, following infection with the GFP tagged BoHV-1 at MOI = 0.1 (top panel, Fig 6D) or MOI = 3 (bottom panel, Fig 6D). While the growth rates in GFP intensity in the rescue cells (VPS52R) were almost identical to that in the control cells (Clone P), there was a profound reduction for the KO cells in the MOI = 0.1 group, as its GFP signal barely increased throughout the infection (Fig 6D), indicating impaired multi-cycle infection dynamics. A decrease in GFP was also observed in the KO cells

within the MOI = 3 group but it was not as dramatic; a 50.1% reduction on average was observed for all KO cells by 48hpi (Fig 6D).

## VPS50,54 double KO but not single KO recapitulates the effects of VPS51, VPS52 and VPS53 KO

Because GARP and EARP share subunits VPS51, VPS52 and VPS53, our data so far indicate that at least one of the complexes is required for efficient BoHV-1 replication. To determine which, we created EARP or GARP specific KO cell lines by deleting just VPS50 (VPS50KO) or VPS54 (VPS54KO) (**Table E in** S2 Text). Interestingly, there was only a small reduction (1.4-fold in VPS50KO, p<0.0001; and 1.6-fold in VPS54KO, p<0.0001) in virus titre compared to the control (Clone P, Fig 6E) when we performed plaque assays on them; and no change in plaque sizes was observed for both series of KOs (**Figs Q and R in** S2 Text). Although still significant, the decrease of virus titre in VPS50KO or VPS54KO cells was modest compared to that observed in VPS51KO, VPS52KO and VPS53KO. These results suggest that GARP and EARP serve alternative roles for BoHV-1 and can be recruited by the virus via the same pathway or independent routes. To test this, we produced double knockout cells lacking both VPS50 and VPS54, i.e. VPS50;54dKO clones (**Table E in** S2 Text) and conducted plaque assays as before. The dKO cell clones recapitulated the results obtained from VPS51KO, VPS52KO and VPS53KO clones with close to 3-log reduction in virus titer (Fig 6E). These experiments suggest that the roles GARP and EARP play are interchangeable for BoHV-1 but that at least one of them is required for efficient BoHV-1 replication.

## GARP and EARP knockout also affects other herpesviruses including HSV-1 and AIHV-1

We then set out to determine whether GARP and EARP deficiency affects other viruses beyond BoHV-1. We first tested this on another alpha-herpesvirus HSV-1, in the human lung carcinoma epithelial cell line A549 cells. Although we were unable to generate bi-allelic 51KO and 52KO A549 cells, haplo-deficiency in mono-allelic KO cells, huVPS51+/- and huVPS52 +/-, was sufficient to lead to a significant albeit modest 2-fold reduction in HSV-1 titer (Fig 6F). Similarly, HSV-1 titer dropped in VPS50KO and VPS54+/- A549 cells (Fig 6F). Future work is needed to confirm the negative impact on HSV-1 replication by generating bi-allelic GARP/EARP knockout human cell lines. We then tested a herpesvirus from a different subfamily other than alpha-herpesviruses. Alcelaphine herpesvirus-1 (AlHV-1) is a gamma-herpesvirus that infects wildebeests and domestic cattle [69]. When we titrated this virus (Strain C500) on the GARP and EARP KO MDBK cells (VPS51KO, VPS52KO, VPS53KO), there was a ~3-log reduction in virus titer (Fig 6G), whereas the reduction in GARP only (VPS54KO) or EARP only (VPS50KO) KO cells was very moderate (Fig 6H). These results were comparable to what was observed for BoHV-1, indicating that AIHV-1 might employ the GARP/EARP complexes in a similar fashion to BoHV-1.

## Initial virus entry and viral gene transcription efficiency are unaltered in VPS52KO cells

To understand the mechanism behind reduced infectibility, we investigated whether the loss of GARP and EARP affected the virus during its earlier stages of infection. Using DNA samples collected from pellets of VPS52KO cells and unedited cells (Clone P) infected with BoHV-1 (MOI = 0.3) as templates, we first quantified the viral genome within these cells post infection across six time points from 0hpi to 24hpi by qPCR. Little difference in viral genomic

DNA (vDNA) copy number was observed between the KO and control cells (Clone P) up to 12hpi, indicating that the loss of VPS52 did not impact on viral entry from the cell surface to the nucleus (Fig 7A). Using RT and qPCR, we then measured mRNA transcription efficiency of the ICP0 (IE), Circ (IE), UL23 (E), and Glycoprotein B (L) genes in these cells. For all four genes examined, there was minimal difference in mRNA/ vDNA ratios between the VPS52KO cells and control cells throughout the course of infection (Fig 7A). This suggests that the loss of the tethering complexes did not affect viral gene transcription efficiency in the VPS52KO cells and therefore may not be the explanation for the observed reduction in viral titer.

## Most viruses produced by VPS52KO cells are not infectious

Given that it was unlikely the VPS52KO affected the virus at the entry and gene transcription stages during the initial infection, we then sought to understand the impairment by examining the viruses themselves produced by the VPS52KO cells. We first titrated the total, cell-free and cell-associated viruses collected from the VPS52KO and control cells on WT MDBKs. Throughout a 36-hour infection with BoHV-1 at MOI = 0.1, the production of infectious total, cell-free (supernatant), and cell associated viruses (pellet) were all greatly impaired in VPS52KO cells (Fig 7B). There was a rate of reduction at ~1-log per cycle compared to non-KO cells (Clone P), and a ~3-log cumulative reduction by 36hpi after about three cycles of replication. Significant differences in virus titers were also observed between the control cells and 52KO cells when they were challenged by the virus at MOI = 3(Fig 7C).

However, when we quantified the virus particles produced by these cells using qPCR, the viral genome copy numbers followed a similar upward trend for both VPS52KO and the unedited cells (Clone P). This is true for most timepoints examined during the 36-hour infection for both MOI = 0.1 (Fig 7B) and MOI = 3 infections (Fig 7C). No substantial difference in genome copies was observed for total, cell-free or cell-associated genomes produced between VPS52KO and Clone P cells (Fig 7B), with a maximum of ~1-log decrease for virions secreted to the supernatant by VPS52KO at 36 h.p.i. When we calculated the genome to pfu ratios, they followed a steady downward trend for viruses produced by unedited cells (Clone P), indicating increasingly efficient production of infectious virus particles once the infection was established (Fig 7B). Interestingly, we observed drastically different trends for virions produced by VPS52KO cells. The genome to pfu ratios remained high for total, cell-free and cell-associated viruses produced throughout the time course of infection, for cells infected at MOI = 0.1 (Fig 7B) and at MOI = 3 (Fig 7C). Taken together, these data further imply that the loss of GARP/ EARP function affected the virus at a later stage of its life cycle, likely during cytoplasmic packaging as suggested by the high genome/pfu ratios of released viruses, leading to the production of poorly infectious viruses.

## Viruses released by VPS52 KO cells severely lacks viral tegument protein VP8

Since GARP and EARP are involved in the trafficking of proteins between membrane vesicles [49], we conjectured that they are employed in the sourcing and trafficking of membrane associated viral proteins that are crucial components of infectious virions. According to this theory, trafficking routes of these viral proteins are cut off in cells missing both complexes during cytoplasmic packaging, resulting in virions deficient in such proteins that are unable to establish new infection. Thus to further understand the invovlement of GARP/GARP, we assayed the protein contents of virions released by these KO cells, with a focus on membrane associated viral proteins in the first instance.

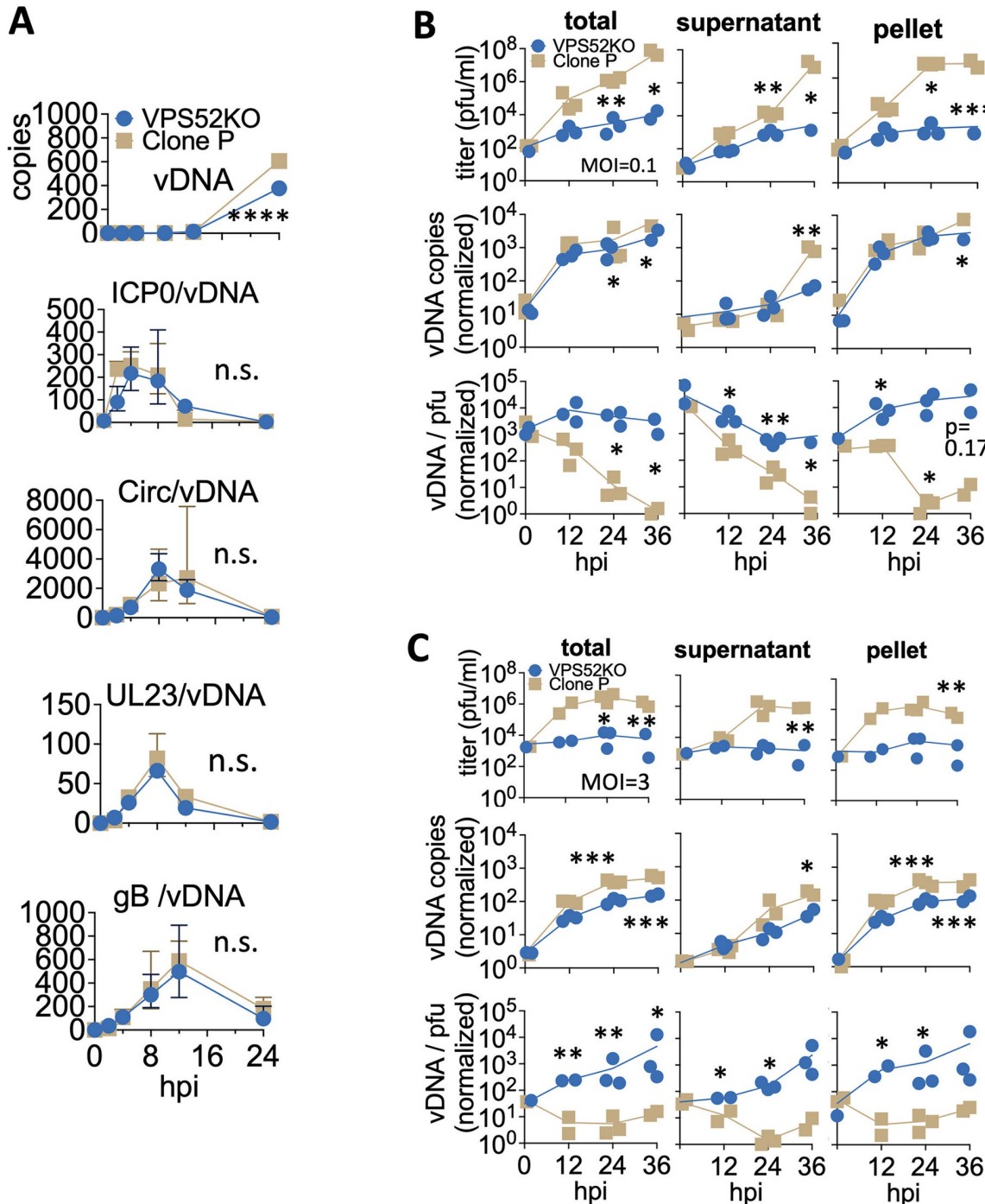

**Fig 7. Dual loss of GARP and EARP did not alter viral gene transcription but lead to higher genome to pfu ratios. A.** Relative viral genome (vDNA) copy numbers in VPS52KO or control (Clone P) cells infected with BoHV-1 at MOI = 0.3 over 24 hours. mRNA/ vDNA ratio were also tracked for immediate early (ICP0, Circ), early (UL23) and late (glycoprotein B or gB) viral transcripts. **B.** Growth curves of titres, viral genome copy numbers (vDNA copies), and genome to pfu ratios (vDNA /pfu) of viruses collected from the supernatant, cell pellet or total (supernatant + cell pellet) samples of VPS52KO or control cells (Clone P) infected with BoHV-1 at MOI = 0.1 over 36 hours, with each dot representing one data point (n> = 2). For the vDNA copies and vDNA/pfu data series, all data points were normalized to the smallest measurements which were set as 1 prior to plotting. **C.** Same experiment as in B but with MOI = 3. Timepoints with significant difference between genotypes are highlighted (****: p<0.0001; ***:0.0001 to 0.001; **:0.001 to 0.01; *: 0.01 to 0.05).

To achieve this, we obtained a monoclonal antibody generated by the Friedrich-Loeffler-Institut (FLI) that targets a viral protein of BoHV1 strain "Schönböken" (https://www.european-virus-archive.com/antibody-or-hybridoma/murine-mab-anti-envelope-protein-bovine-herpesvirus-1-bohv-1) [70], which is closely related to strain "Jura" used to generate the GFP tagged BoHV-1 for our screens. Although the antibody target was classified as an envelope protein back in 2005 when it was first produced (personal communication with Dr. Sven Reiche of FLI), our Mass Spectrometry and western blot data demonstrate that it specifically binds to VP8 of BoHV-1 strain "Jura" (**Fig S in** S2 Text). As the most abundant tegument protein of BoHV-1 [71], VP8 is a major component of the outer tegument layer of BoHV-1. During the late stage of infection, it is translocated from the nucleus to the Golgi and cytoplasm where it is packaged into maturing viruses prior to cellular egress [72].

Using this antibody, we first determined whether the expression level and overall distribution of VP8 was affected in the VPS52KO cells. By immunofluorescent staining and confocal microscopy, we detected ample VP8 proteins in the VPS52KO cells at 10 h.p.i. with BoHV-1 at MOI = 1. This is comparable to that in the control cells (wt and Clone P, Fig 8A), indicating that the loss of GARP and EARP did not affect VP8 expression. We also did not observe any substantial difference in VP8 protein distribution relative to a cis Golgi marker GM-130 (Fig 8A). However, due to the lack of antibodies against other organelles, such as the Endoplasmic reticulum and endosomes that work properly in the MDBK cells, any more specific changes in the VP8 localization within the VPS52KO cells remain to be further explored. By confocal microscopy we also failed to detect any apparent alterations in VP26-GFP protein expression in the KO cells (Fig 8A).

Given that the GARP/EARP KO unlikely altered viral protein expression, we proceeded with experiments to determined if it affected viral protein inclusion into virions instead. To assay this we probed VP8 composition of the cell free viruses released by the KO cells, with the level of VP26-GFP as a proxy for the quantity of virus particles. By infecting the KO cells with the BoHV-1 GFP virus at MOI = 3, we harvested released viruses via ultracentrifugation of the tissue culture supernatants at 48h.p.i. (Fig 8B). Upon lysing the virus pellets under reducing conditions, we conducted western blot analyses by hybridizing the samples with the anti-VP8 antibody (Fig 8A) and an anti-GFP antibody. Compared to viruses released by the WT and Cas9+ MDBK (Clone P) cells, we observed drastic decreases of VP8 content in viruses produced by the VPS52KO cells. The ratio of VP8 to GFP was reduced by >90% compared to that of the control cells (Fig 8C and 8D). This is in sharp contrast to the 2-log increases in genome to pfu ratios of the viruses produced by the same cells (Fig 8E).

Although by confocal microscopy we didn't observe any obvious impairment to the VP8 protein expression in the infected VPS52KO cells (Fig 8A), to further rule out the possibility that it is majorly responsible for the decrease of VP8 on released viruses, we quantified the VP8 and VP26-GFP proteins in the cells by western blot. We found that the VP8 protein levels stayed the same between the KO and non-KO cells (Fig 8F and 8G) while the VP26-GFP decreased by 3-fold (Fig 8H). Interestingly and contrary to the reduced VP8 level on released viruses (Fig 8C and 8D), the VP8 to GFP ratio in the VPS52KO cells doubled compared to the control cells (Fig 8I), suggesting retention of VP8 in the KO cells as less of it was included into viruses that were subsequently released into the extracellular space.

## Discussion

By constructing and using a bovine genome wide CRISPR knockout library, we were able to conduct genome wide screens during BoHV-1 infection. As a result, we produced a list of proviral and anti-viral host factors that could impact various aspects of BoHV-1 biology such as

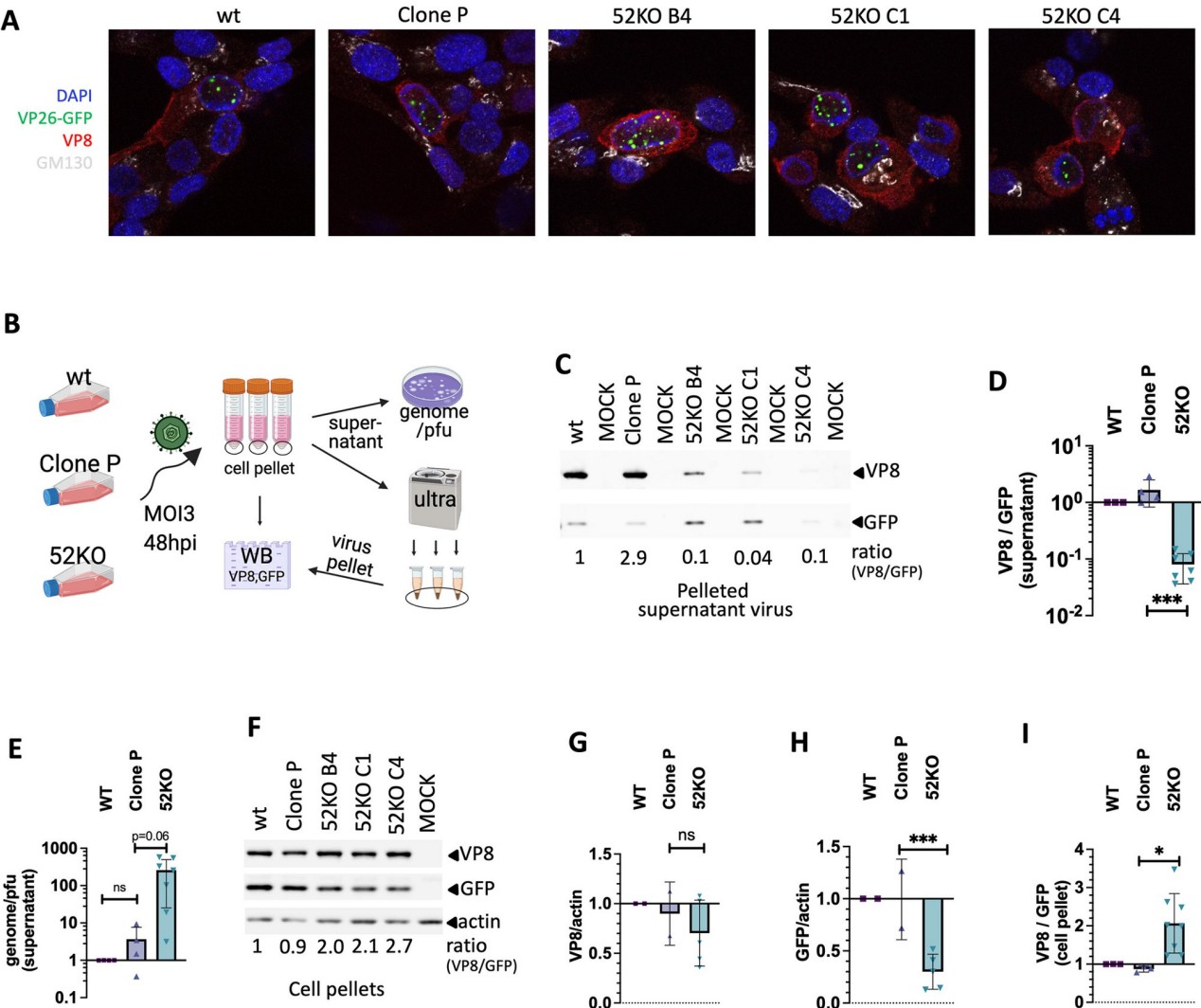

**Fig 8. VPS52 KO reduces VP26-GFP in cells and the amount of VP8 present on released cell free virions. A.** Confocal imaging of WT, Clone P (Cas9+/+), and three clones of VPS52KO MDBKs, infected with the GFP tagged BoHV-1 virus and stained with DAPI(nucleus), GFP(VP26), Alexa 594 (red for VP8) and Alexa 647 (white for cis Golgi marker GM130) at 10hpi. **B.** Schematic drawing of the experimental setup to quantify the VP8 and GFP proteins. Same cells used in A infected with BoHV1-GFP at MOI = 3 and incubated for 48 hours. A small portion from each tissue culture supernatant was used for plaque assays and qPCR to obtain genome/pfu ratios with the bulk concentrated by ultra-centrifugation. The virus pellets were lysed along with the cell pellets and all lysates were then assayed by Western Blotting using antibodies against VP8, GFP, and beta actin (cell lysates only). Figure drawn with Biorender (https://www.biorender.com/). **C.** Sample Western Blot results of pelleted viruses from the supernatants. The ratio of VP8 to GFP banding intensity measured by densitometry is shown at the bottom for each sample after normalization to that of the WT (set at 1). Lysates for viruses collected from the WT and Clone P cells were diluted 10 times with Laemilli buffer before 2ul of each sample were loaded to avoid signal saturation. **D.** Bar chart of VP8 to GFP ratios of concentrated cell free viruses from each cell line measured as in **C** (***: p<0.001, n = 3). **E.** genome to pfu ratios of cell free viruses calculated as in Fig 7B and 7C (p = 0.06, n = 3). **F.** Sample Western Blot results of cell lysates collected as in **B** blotted with antibodies against VP8, GFP and beta actin. VP8 to GFP ratios for all cell lysates were calculated as in C and normalized to that of the WT cells. **G,H,I.** Bar charts of VP8 to actin, GFP to actin, and VP8 to GFP ratios for each cell lysate normalized to that of WT cells based on repeated experiments (***: p<0.001; *:p<0.05, n.s.: p>0.05; n = 2).

entry, viral genome replication, viral gene transcription, and intracellular trafficking of viral components. Some of the most intriguing examples include the EARP and GARP complexes, known for promoting endosome recycling [49]. Our follow up experiments suggest that for alpha- and gamma-herpesviruses these complexes serve as crucial links for the trafficking of

viral proteins to the sites of cytoplasmic envelopment. The discovery of multiple other candidate genes involved in retrograde trafficking from EE to the TGN in addition to subunits of GARP and EARP (Fig 6A), strengthens this proposition. Prior to our screen, GARP has been shown to be important for the envelopment of poxviruses, including the vaccinia virus and Monkeypox virus [73,74]. To our knowledge, this report is the first documentation on their requirement for herpesviruses and provides evidence that both the TGN and REs can be the membrane source for envelopement.

Initially these complexes were identified by the screens designed to enrich for cells with reduced or enhanced VP26-GFP protein expression. Indeed, in GARP and EARP deficient cells we observed up to ~3 fold decrease in VP26-GFP protein levels in our VP26-GFP tracking expriments (bottom panel, Fig 6D), and subsequently by Western Blotting (Fig 8H). However, seeing that this 3-fold decrease in VP26-GFP was inadequate to explain the 3-log reductions in virus titers (Figs 6B, 7B and 7C), we conducted additional experiments to understand if the loss of GARP and EARP resulted in other impairments. We found that these deficient cells can release new viruses almost as efficiently as WT cells (Fig 7B and 7C). However, most of the virus particles from the KO cells were not able to establish new infection, demonstrated by their high genome to pfu ratios (Figs 7B, 7C and 8E), indicating that they contain defects that render them non-infectious.

To identify such defects, we examined the protein content of these viruses and discovered that compared to viruses produced by WT cells, viruses released by the GARP and EARP KO cells severely lack VP8 (Fig 8C and 8D). Encoded by the $U_L47$ gene of BoHV-1, the multi-functional tegument protein VP8 is crucial for BoHV-1 virulence. Analogous to VP13/14 from HSV-1, VP8 is a powerful interferon antagonist [75–77] and has also been shown to regulate host cell apoptosis [78]. Viruses harvested from a $\Delta U_L47$ mutant lacking VP8 displayed severe growth defects in tissue culture conditions [79]; both intracellular and extracellular mutant viruses exhibited more than 100-fold decrease in titer, comparable to our viruses obtained from the KO cells (Figs 6B, 6C, 7B and 7C). These findings strongly suggest that the 90% reduction in VP8 tegument protein (Fig 8C and 8D) was a direct cause for the lost infectivity of viruses produced by these KO cells. This reduction was unlikely due to a major decrease in VP8 protein expression as our experiments failed to detect any such change in the KO cells (Fig 8A, 8F, 8G and 8I). All in all, our data provide evidence that the loss of GARP and EARP manifested its negative impact on the virus through disruption to the VP8 packaging in the cytoplasm.

How could these membrane tethering complexes be involved in the packaging of VP8? Previously, Schindler et al [49] demonstrated that GARP is primarily found on the TGN while EARP mainly resides on REs, potentially mediating fusion of vesicles derived from early endosome (EE) with both the TGN and REs. Vesicles of both TGN [80] and RE [22] origin have been strongly indicated as assembly sites for cytoplasmic envelopment of BoHV-1, although this is a heavily debated topic [17,18]. Alpha-herpesviruses including BoHV-1 express multiple membrane bound glycoproteins that are packaged into virions during cytoplasmic envelopment [81]. Prior to packaging, they are concentrated to the plasma membrane by secretion and recycled back to the cytoplasm via endocytosis, with some of them localized to REs [17,18] or vesicles positive for TGN46, a Trans Golgi Network (TGN) marker, such as gH [17,18] and gM of HSV-1 [82]. Thus it is likely GARP and EARP KO disrupted the trafficking of viral glycoproteins from the plasma membrane to the REs and TGN. Could this be related to the packaging defect seen with VP8?

Published research suggests so. Recently a glycoprotein M (gM) deletion mutant of BoHV-1 lead to cell-free viruses with significantly reduced VP8 content and more than 2-log loss in virus titer [72]. The study also demonstrated that for the WT virus, VP8 colocalized itself with

gM on the Golgi after nuclear export. For the gM deletion mutant however, the Golgi localization of VP8 was lost, suggesting the importance of gM for the correct localization of VP8, likely due to the observed interaction between the two [72]. The gM of HSV-1 has been shown to be crucial for the localization of multiple tegument proteins to the assembly sites, including VP13/14 and other envelope proteins, as well as their eventual packaging into virions [82]. Given that the gM protein is conserved across the Herpesviridae species, this could also be true for gM of BoHV-1. In conclusion, these data suggest that GARP and EARP KO might have affected the packaging of VP8 via their trafficking of gM, its interactive binding partner, to the site of secondary envelopement.

Based on our data the literature presented above [17,18,49], we developed a model for this process. Under normal conditions (Fig 9), GARP and EARP serve as key links for vesicles from early endosomes (EE) to be tethered to the TGN and fast REs, allowing membrane fusion to take place and relaying gM and the attached VP8 to these new organelles. Vesicles loaded with gM and VP8 then bud from the TGN and REs, taking gM and VP8 to the destinations where secondary envelopment of the viral capsid takes place. Alternatively, endocytosed gM can be trafficked via slow REs, presumably less efficiently. In cells missing GARP and EARP, gM carrying vesicles are unable to be tethered to the TGN or fast REs, depriving the capsid access to gM-containing vesicles for envelopment as a result. Instead, the capsid resorts to vesicles lacking gM and VP8, producing virions deficient in these proteins that are unfit for

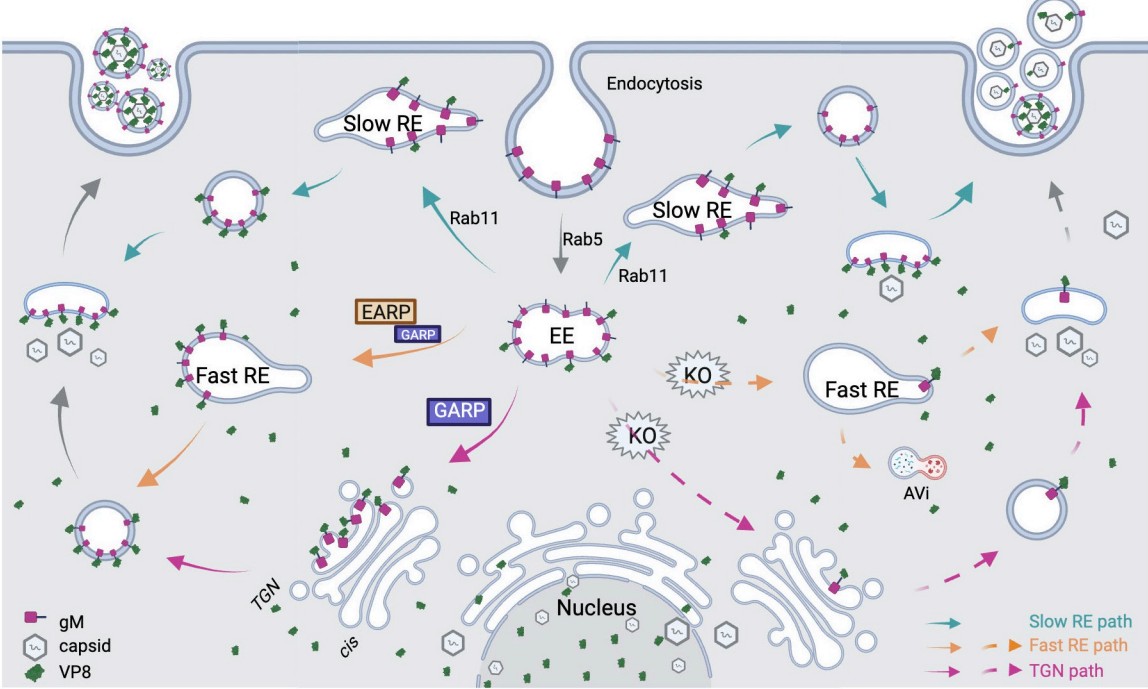

**Fig 9. A simplified model for the trafficking and envelopement of viral proteins of BoHV-1, with a focus on gM and VP8.** Other viral proteins and potential interactions are not depicted for clarity. Viral glycoprotein M (gM, red spikes) are endocytosed from the cell membrane and trafficked to four destined organelles: 1. Slow RE via Rab11 (green paths), 2. GARP/EARP mediated Fast RE (oragne paths), 3. GARP tethered retrograde to the TGN (purple paths), or 4. degradation down the LE path (not shown). The trafficking of viral gM to Fast RE and the TGN is cut-off by GARP and EARP KO (dashed arrows), and vesicles that bud off them are deficient in viral glycoproteins and are primarily used for secondary envelopment of capsids than the Slow RE route, producing virions lacking essential glycoproteins for cell entry. AVi: initial autophagic vacuoles, EE: early endosomes; RE: recycling endosomes; TGN: trans-Golgi network. Figure drawn with Biorender (https://www.biorender.com/).

infection (Fig 9). EEs carrying gM are more likely to be diverted to late endosomes or initial autophagic vacuoles (AVis) for degradation. Some capsids can still mature into infectious virions by using vesicles from slow REs instead. Based on the ~1-log / replication cycle reduction of infectious viruses produced by VPS52KO cells (Fig 7B), we estimate it to be less than 10% as efficient as the GARP/EARP mediated paths. Another possibility is that capsids without a secondary envelope and the glycoproteins are released into the culture media at the end of the infection cycle via host cell apoptosis, but this remains to be examined.

A good understanding of host pathogen interaction in livestock is crucial for animal health and the control of zoonotic diseases. Like other herpesviruses, BoHV-1 has co-evolved to tightly regulate and usurp the host cell niche. As many of the pro- and anti-viral candidates identified by our CRISPR screen remain to be studied, we hope that further dissection of the roles these genes play in viral propagation may aid development of therapeutics. Among them, candidate genes that are also core essential genes (CEGs) [44] or fitness genes might not be as straightforward to study using the CRISPR KO approach, as they are essential for the growth of cells in prolonged tissue culture conditions. Due to the lack of a cell- or tissue-based latency reactivation model, we were confined to the study of the active cycle of BoHV-1; it may be beneficial to adapt this screen to gain knowledge regarding the maintenance of BoHV-1 latency when the opportunities arise. Besides BoHV-1, the screening platform we built is readily deployable to the study of other economically important bovine pathogens, such as foot and mouth disease virus, bovine rotavirus, bovine viral diarrhoea virus, bovine respiratory syncytial virus and Schmallenberg virus. One can also use this library to study aspects of biology beyond virology, such as pluripotency maintenance and differentiation of stem cells and repurposing BoHV-1 as a cancer therapy vector [83–86].

## Materials and methods

Briefly, a library of 94,000 CRISPR sequences targeting 21,165 protein coding genes, along with 2,000 non-targeting CRISPRs was synthesized as 80-mer single strand oligos by Twist Biosciences (San Francisco, USA) in the following format: GCAGATGGCTCTTTGTCCTA-GACATCGAAGACAACACCG-N20-GTTTTACAGTCTTCTCGTCGC (N20: 96,000 variable 20bp CRISPR sequences). The oligos were then PCR converted to dsDNA and cloned into pKLV2-U6gRNA5(BbsI)-PGKpuro2ABFP-W [39] by BbsI digestion and T4 ligation to make the K2g5 library. The other three libraries were cloned using the same protocol but with different vector backbones. The library was packaged into lentivirus in HEK293FT cells by Calcium Phosphate transfection and transduced into ~100 million Cas9+/+; TRIM5 -/- cells at a MOI of 0.3–0.5 with three repeats. Transduced cells were selected with 1.8 ug/ml Puromycin for 7–10 days, while maintaining a coverage of 300-500x. For both screens, 70–100 million (passage 4–5) library transduced cells were infected with GFP tagged BoHV-1 virus at a MOI of 2. At 10hpi (1st screen) or 8phi (2nd screen), cells were harvested for FACS sort, to obtain fractions of live cells with varied GFP intensity, i.e. GFP negative, GFP low, GFP medium, and GFP High. The viral infection was repeated three times and genomic DNA was extracted from all fractions and all repeats. The CRISPR containing lentiviral sequences were amplified and barcoded by a 2-step PCR with limited cycles and sequenced on a NextSeq 500 machine using a SR75bp High Output kit. Reads for each sample were then trimmed using cutadapt v1.16 [87] and counted based on the CRISPR sequences they contain using MAGeCK v0.5.8 [43]. Pairwise comparisons of CRISPR copy numbers between fractions were also completed by MAGeCK, to identify genes with significantly enriched or depleted guides in different fractions post BoHV-1 infection. For more details such as PCR primers, CRISPR and TALEN sequences, plasmids and reagents used in this study, please refer to the S1 Text.

## Supporting information

**S1 Data. btCRISPRko_v1_library.**
(XLSX)

**S2 Data. g2 vs g5.**
(XLSX)

**S3 Data. 1st_screen_raw_counts_and_MAGeCK_results.**
(XLSX)

**S4 Data. 2nd_screen_raw_counts_and_MAGeCK_results.**
(XLSX)

**S5 Data. All_candidates.**
(XLSX)

**S6 Data. focused_screen_related_info.**
(XLSX)

**S7 Data. GO_analyis_of_all_candidates.**
(XLSX)

**S1 Text. Supplementary_materials_and_methods.**
(DOCX)

**S2 Text. Supplementary data.** Table A. Numbers of genes targeted, and guides included in the btCRISPRko.v1 library. Table B. Libraries produced from this study. Table C. Number of cells recovered from 1st screen. Table D. Number of cells recovered from 2nd screen. Table E. Knockout clone genotypes. **Fig A.** Knock-in (KI) of Cas9 to rosa26. **A.** Knock-in and genotyping strategies. A genome editor designed to target first intron of rosa26 generates a double strand break and a plasmid with a EF1a promoter driven Cas9 expression cassette flanked by homology sequences (orange colored) is used to repair the break, incorporating the expression cassette. To identify targeted clones, PCR primer set F+R binding outside the homology arms (horizontal read arrows) is used for genotyping. **B.** TALEN mRNA pair TAL1.6, chosen out of six editors tested (T7E1 results shown on gel), cuts in intron 1 and was co-transfected with HDR plasmid expressing the Cas9-2a-Blasticidin cassette. Single cell clones were isolated and genotyped. **C.** Genotyping results. PCR amplicons from the 4kb Wild Type (wt, filled triangles) and 11kb targeted (tg, filled triangle) alleles are indicated. Homozygote clones are labelled as +/+. **Fig B. Lentivirus transduction efficiency is low in wt MDBK cells.** A. bright field visualization of HEK293FT cells 48 hours after being transfected with the library plasmid pool, pMD.2 and psPAX2 for serum-free lenti-virus packing. B. Same cells under the BFP filter showing strong BFP expression, indicating efficient transfection and packaging. C. CRISPR library transduction in HEK293FT cells. D. CRISPR library transduction in Cas9+/+ cells without TRIM5 KO with four times the virus as that used on HEK293FTs. **Fig C. Plaque assays to compare replication of BoHV-1 in wild type MDBK cells and homozygous Cas9 clones.** A. Plaque sizes measurements from wt, and three Cas9$^{+/+}$ clones infected with GFP tagged BoHV-1 virus, repeat n = 3 (n.s.: not significant with p-value>0.05 based on a two-tailed t-test). B. Plaque number counts calculated as pfu/mL from wt, and the Cas9$^{+/+}$ clones. **Fig D. Plaque assays to compare replication of BoHV-1 in wild type MDBK cells and TRIM5-/- clones.** A. Plaque sizes measurements from wt, and two TRIM5 KO clones, A44 and B13 infected with GFP tagged BoHV-1 virus, repeat n = 2. B. Plaque number counts calculated as pfu/mL from wt, the TRIM5 KO clones. C. Sample plaque assay results with the

same quantities of virus. **Fig E. Plaque assay to compare BoHV-1 replication in Cas9 +/+ only cells and Cas9+/+; TRIM5-/- clones.** A. Titer of virus grown in Cas9+/+ or Cas9+/+; TRIM5-/- cells; B. Average size of plaques grown. All comparisons conducted by single factor ANOVA results non-significant results. **Fig F. Testing CRISPRs designed for the CRISPRko library in MDBK cells.** A. Cutting efficiency of guides delivered by lentivirus or PiggyBac targeting CTNNB1, MAPK8 and MAPK9. B. Cutting efficiency of guides *in vitro* transcribed as sgRNA and delivered by transfection, targeting Oct1, SGK1, SGK2, and SGK3. **Fig G. Stepwise CRISPR library cloning, packaging, and transduction. The l**ibrary was synthesized as a pool of oligos and PCR amplified to convert to dsDNA. The fragments containing CRISPRs were then ligated into either lentivirus (illustrated as example) or piggyBac vectors that contain a hU6 promoter to drive sgRNA expression and a Puro2aBFP marker for Puromycin selection. If using lentiviral delivery, the library was packaged as lentivirus and transduced into Cas9 expressing cells at low MOI to produce library expressing cells with single sgRNA integrations. **Fig H. Next generation sequencing of screening samples to identify candidate genes with depleted or enriched guide RNA.** To determine copy numbers of guides, PCR using primers PCR1_Fx + PCR1_R (red arrows, Table E in S1 Text) are used to amplify the fragments containing CRISPR sequences. PCR products from different samples are then barcoded using a 2nd PCR with a unique combination of indexes and sequenced as a pool. Comparison of guide RNA copy numbers between samples is conducted by MAGeCK to identify enriched or depleted guides and their targeted genes. **Fig I. NextSeq of the CRISPR libraries to examine CRISPR copy number distribution.** A. Sequencing of the K2g2 and K2g5 plasmid libraries at 40X sequencing depth. B. Sequencing of the K2g5 library and library transduced Cas9+/+; TRIM5-/- cells at 200X sequencing depth. **Fig J. Library performance and specificity comparison between the K2g2 and K2g5 libraries.** Plots show log2 fold changes in copy numbers of guides targeting essential genes(red), non-essential genes(black), and non-targeting control guides(green) in the two cell populations transduced with the K2g2 or K2g5 library compared to the plasmids. The data was plotted using a python script adapted from Hart et al. 2017 [44]. **Fig K. PiggyBac transposition mediated CRISPR library delivery into Cas9+/+ MDBK cells.** Cas9+/+ transfected with either library with (left panels) or without (right panels) transposase hypBase and selected with Puromycin to test library transposability. After Puro selection, cells were stained with Giemsa for colony visualization. **Fig L. GFP tagged BoHV-1 virus infecting wt MDBK cells.** A. 8 hours post infection of MDBKs by the GFP tagged virus. B. image of the same cells under bright field. **Fig M. FACS sort to collect sub-populations of different degrees of BoHV-1 replication.** FACS plots with four gatings to collect sub-populations of live cells with GFP Negative, GFP Low, GFP Medium, and GFP High signals from 1st or 2nd round of screen. **Fig N. Gene Ontology analysis of pro-viral candidates identified by the screen.** GO reveals the top biological processes that can be important for the virus using the Fold Enrichment $>$ = 10, and -log2(p-value)$>$ = 10 statistical cut-offs (To see the full list, please refer to S6 Data). **Fig O. Candidates identified in the GFP Negative sub-population.** Data shown is from 2nd screen, labels only applying to pro-viral candidates with stringent cut-offs: p$<$0.005, FDR$<$ = 0.1, and log2fc$>$ = 1.5 (S3 Data). **Fig P. Candidates identified in the GFP Low sub-population.** Data shown is from 2nd screen, labels only applying to pro-viral candidates with stringent cut-offs: p$<$0.005, FDR$<$ = 0.1, and log2fc$>$ = 1.5 (S3 Data). **Fig Q. Sizes of BoHV-1 plaques grown in various clones.** ImageJ was using to measure sizes of plaques grown in VPS50KO, VPS51KO, VPS52KO, VPS53KO and VPS54KO (denoted as -/-) cells as well as KO clones with overexpression of corresponding subunit (pVPS50,51,52,53,54). Cas9+/+ clone P was used as control and plaque size for the control was normalized to 1 (n.s.: p$>$0.05 based on ANOVA followed by t-tests between genotypes). **Fig R. Plaque size in VPS50KO and VPS54KO cells.** Plaques were grown and measured in Control cells (Clone P),

VPS50KO (left panel) clones or VPS54KO clones (right panel). Average plaque size from Clone P control cells were set as 1 (n.s.: p>0.05 based on ANOVA followed by t-tests between genotypes). **Fig S. Specificity of an antibody against VP8 of BoHV-1.** A. Western blot result using the anti-BoHV-1 antibody (red) and an anti-GFP antibody (yellow) against a concentrated virus sample harvested at 48h.p.i. (MOI = 3) from tissue culture supernatant of WT cells grown in serum free media (SFM, DMEM+1% Pen/Strep). Black arrows point to bands that match the size of VP26-GFP (41kD) and the most prominent band on the membrane, marked with a "?" as the candidate target antigen for the FLI antibody. SFM was used as a negative control. B. Concentrated virus samples resolved on 10% SDS PAGE gels and stained by Coomassie. PAGE gel pieces (marked by the black rectangular boxes) containing bands that match the size of the detected candidate band by western blot in A were cut out and protein contents were examined by Mass Spectrometry. C. Proteins with the most abundant peptides in viruses harvested from the WT cells and the MOCK samples detected by the Mass Spectrometry (n = 2). The Y-axis represents abundance of each protein as total area under curves of all peptides returned for that protein. D. Representative Western blot results on lysates harvested from HEK293FT transfected with plasmids overexpressing glycoproteins or VP8 of BoHV-1. The same samples were hybridized with the primary anti-Flag antibody (rabbit) and the FLI antibody (mouse) together followed by secondary antibodies labelling the anti-Flag (Licor Donkey anti-rabbit 800) or the FLI antibody (Licor Donkey anti-mouse 680). -ve: empty vector, MOCK: lysate from supernatant of MOCK infected cells; BoHV-1: lysate from concentrated supernatant of cells infected with the GFP tagged BoHV-1.
(DOCX)

## Acknowledgments

We thank Dr. Peter Wild and colleagues at ZTH for sharing the GFP tagged BoHV-1 strain, Prof. Dr. Jens Teifke and Dr. Sven Reiche for sharing the anti BoHV-1 antibody, Dr. Kosuke Yusa from the Wellcome Sanger Institute, Amy Tong from Professor Jason Moffat's lab at the University of Toronto, and Professor Keisuke Kaji from the SCRM Medicine at the University of Edinburgh for sharing their protocols, and other members of the CRISPR community for wonderful insights and techniques. We are also grateful to the imaging facility staff Dr. Anna Raper, Dr. Bob Fleming and Mr. Graeme Robertson and the CSU unit for support, the virology group for meaningful discussions, Professor Kenneth Baillie, Professor Mike McGrew, Dr. Lel Eory, Dr. Oriol Xandri Canela, Dr. Mazdak Salavati and Dr. Rute Maria Pinto for advice or comments on the manuscript, all based at the Roslin Institute. We would also like to extend our appreciation to the NGS team, Angie Fawkes, Richard Clark and Lee Murphy at the ECRF for sequencing our screen samples.

## Author Contributions

**Conceptualization:** Wenfang S. Tan, Simon G. Lillico, Bruce Whitelaw, Robert G. Dalziel.

**Data curation:** Wenfang S. Tan.

**Formal analysis:** Wenfang S. Tan, Paul Digard.

**Funding acquisition:** Wenfang S. Tan, Simon G. Lillico, Andy Law, Paul Digard, Bruce Whitelaw, Robert G. Dalziel.

**Investigation:** Wenfang S. Tan, Enguang Rong.

**Methodology:** Wenfang S. Tan, Enguang Rong, Paul Digard.

**Project administration:** Wenfang S. Tan, Robert G. Dalziel.

**Resources:** Wenfang S. Tan, Enguang Rong, Inga Dry, Andy Law, Paul Digard.

**Software:** Wenfang S. Tan.

**Supervision:** Wenfang S. Tan, Robert G. Dalziel.

**Validation:** Wenfang S. Tan, Enguang Rong.

**Visualization:** Wenfang S. Tan.

**Writing – original draft:** Wenfang S. Tan.

**Writing – review & editing:** Wenfang S. Tan, Inga Dry, Simon G. Lillico, Paul Digard, Robert G. Dalziel.

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
