## [Decision Letter · Decision Letter 0]

9 Jan 2023

Dear Dr. Tan,

Thank you very much for submitting your manuscript "GARP and EARP are required for efficient BHV-1 replication as identified by a genome wide CRISPR knockout screen" for consideration at PLOS Pathogens. As with all papers reviewed by the journal, your manuscript was reviewed by members of the editorial board and by several independent reviewers. In light of the reviews (below this email), we would like to invite the resubmission of a significantly-revised version that takes into account the reviewers' comments.

Each of the three reviewers appreciated a number of strengths of the manuscript, including the construction of a useful resource for the field and its application in a CRISPR screen of BHV1 replication. Each of the three reviewers make points that will be important to address in a revision. One of the reviewers inadvertently typed these points in the wrong field, so including them here:

1. Bovine herpesvirus 1 is now abbreviated as BoHV-1, not BHV-1. 2. GARP and EARP were not defined in the abstract. 3. Is it accurate to state that GARP and EARP are redundant functions that BoHV-1 needs for retrograde transport and virion production?

4. One major, but unavoidable, flaw with this approach is cellular genes necessary for cell growth will not be viable and thus cannot be tested. This should be discussed in the manuscript. 5. Figure 6F: the differences were less than one log in virus production. Are the authors sure this difference is biologically relevant?

We cannot make any decision about publication until we have seen the revised manuscript and your response to the reviewers' comments. Your revised manuscript is also likely to be sent to reviewers for further evaluation.

Sincerely,

Benjamin E Gewurz, M.D., Ph.D.

Academic Editor

PLOS Pathogens

Patrick Hearing

Section Editor

PLOS Pathogens

Kasturi Haldar

Editor-in-Chief

PLOS Pathogens

orcid.org/0000-0001-5065-158X

Michael Malim

Editor-in-Chief

PLOS Pathogens

orcid.org/0000-0002-7699-2064

Each of the three reviewers appreciated a number of strengths of the manuscript, including the construction of a useful resource for the field and its application in a CRISPR screen of BHV1 replication. Each of the three reviewers make points that will be important to address in a revision. One of the reviewers inadvertently typed these points in the wrong field, so including them here:

1. Bovine herpesvirus 1 is now abbreviated as BoHV-1, not BHV-1. 2. GARP and EARP were not defined in the abstract. 3. Is it accurate to state that GARP and EARP are redundant functions that BoHV-1 needs for retrograde transport and virion production?

4. One major, but unavoidable, flaw with this approach is cellular genes necessary for cell growth will not be viable and thus cannot be tested. This should be discussed in the manuscript. 5. Figure 6F: the differences were less than one log in virus production. Are the authors sure this difference is biologically relevant?

Reviewer's Responses to Questions

**Part I - Summary**

Reviewer #1: For this study, the author utilized a genome wide CRISPR knockout strategy to target all protein coding genes in the bovine genome. As expected, this strategy identified numerous genes that impaired or stimulated viral replication. Interestingly, the Golgi-associated retrograde protein (GARP) complex and endosome associated recycling protein (EARP) complex scored as being very important for virus production. BoHV-1, like most herpesviruses undergoes a complicated strategy to generate infectious viruses that are shed or spread via cell to cell. Hence, it is expected these pathways would be important. With that said, there are several issues that need to be addressed.

Reviewer #2: This manuscript by Tan et al. illustrates the careful construction and impressive documentation of a CRISPR/Cas9 knockout library for high throughput genomics and gene discovery in the cattle genome. The btCRISPR library generated, targeting all protein coding genes in the cattle genome, will be hugely useful in studying cellular processes in bovine cells, as well as processes that intersect with viruses infect bovine cells. At every barrier, the authors went the extra mile to create a high-standard resource. The authors perform extensive gene ontogeny analysis, and illustrate the possible involvement of many pro- and anti-viral host genes involved in the viral replication cycle. They validate one of these sets of genes, comprised of VPS 50-54, which assemble to form the GARP and EARP complexes. They then rigorously knock out the components of these complexes to ascertain their role in the BHV1 life cycle. They arrive at an intriguing model that trafficking of viral glycoproteins to RE and the TGN is cut off by GARP and EARP KOs, and the vesicles that bud off are deficient in viral glycoproteins essential for 2ndary envelopment of capsids. This hypothesis is intriguing and prompts some testable hypotheses.

Reviewer #3: Tan et al., develop tools for and then perform a screen for host cell gene products that affect expression of one of the BHV-1 viral capsid proteins, VP26, during a single cycle of infection (i.e., after a relatively high m.o.i. infection at times prior to completion of a first cycle of infection). They identify many genes with both positive or negative effects on VP26 expression and show nicely that knock-down or CRISPRi inhibition of expression putative proviral host factors affects initiation of plaque formation by BHV-1. This appears to have been a fruitful screen and the cell line and library tools that they have constructed represent a highly useful and significant tool for identifying pro- and anti-viral factors for BHV-1 expression.

The paper then takes a confusing turn as the authors focus on two membrane trafficking-related protein complexes, EARP and GARP and attempt to build a case that these are likely to be important for virus assembly. This is confusing in part because their screen was not structured to find cellular factors that promote events in virus replication past the expression of VP26. Indeed, it is difficult to see how factors that might even be absolutely essential for assembly would have had any effect on the outcome of their screen. Perhaps unsurprisingly, the evidence presented that these factors directly affect assembly is incomplete and not easily interpretable.

The authors show that individual knock-outs of three of three subunits common to the EARP and GARP complexes result in strong inhibition of the ability of BHV to initiate plaque formation and have a small effect on the sizes of the plaques that do form (although this latter measurement is not controlled by rescue of expression). They also attempt to show that these knockouts affect virus production in Figure 6D. The problem here is that the experiments shown in Figure 6D do not measure virus production. Rather, they show kinetics of VP26 expression. The lower graph is particularly telling in that it measures VP26 expression in a single cycle of infection (i.e., the vast majority of cells infected at m.o.i. = 3). This result strongly suggests, despite what is shown in Figure 7, that the primary defect in their VPS51/52/53 knockouts is a strong inhibition of expression of at least some late genes. This is, of course, consistent with the results of the screen in which these candidates were identified. The authors show in Figure 7 that the virions produced in the VPS52 knockout have a lower specific infectivity than those from a wild-type cell clone. This can easily be explained as an indirect effect of inhibition of expression of critical virion components. It should also be noted that the experiments shown in Figure 7 are also not controlled with cells in which the knockout is repaired.

The approaches shown in this manuscript for library generation screening and validation are admirably state-of-the-art. However, to draw the conclusion that the EARP and GARP complexes are directly important in virus assembly, the authors will need to incorporate additional approaches. They must, first, assess the expression of viral proteins (in addition to mRNA) to determine the effects of their knock-outs on expression. They must also perform actual virus production assays (i.e., single- and multi-step growth experiments that measure production of infectious virus rather than marker gene expression). Also, there is no TEM or imaging of any kind to support an effect on virus assembly.

More minor points include:

Figure 1B assay is not explained. This is presumably output from a T7E1 assay, but the basis of the assay as applied in this circumstance is unexplained. Insufficient information is provided to allow critical evaluation of the data. And this is a problem throughout the portion of the paper that describes cell line and library engineering.

Also, no validation of knock-out of TRIM5alpha expression is presented. The authors get the effect they want with some clones, but whether this is related to TRIM5 KO is not clear. In one respect, it does not matter, since they get clones that show elevated transduction efficiency, but the lack of transparency and rigor in characterization of those clones is disturbing.

Given that the screen was based on diminished or enhanced GFP expression in a single cycle of virus replication, how do the authors imagine, they identified genes that are involved in virus assembly and virion specific infectivity?

Table S1 – the meaning of the numbers in the right-hand column is not explained. The table is unintelligible.

Figures S3-S5 do not measure the parameter relevant to their screen. They look, on the one hand, at ability to initiate a plaque and, on the other, at the ability to spread. This is a problem because their screen is based on inhibition or enhancement of late protein expression during a single cycle of infection. The relationship between plaque formation, spread and virus single-cycle replication is complicated (at least in the alphaherpesviruses) and the authors should validate the utility of their cell lines using the same sort of criteria they will use for their library screen.

Figure 2C axes are uninterpretable. Y axis: “Density” of what? What are the units? X axis: is this nucleotides?

**Part II – Major Issues: Key Experiments Required for Acceptance**

Reviewer #1: 1. This manuscript contains 16 supplementary Figures. The authors need to include tables or a couple of figures that summarizes what the top 10 genes or signaling pathways that were identified and include it in the manuscript. Furthermore, these genes need to be named and not just listed as number, and a brief description included.

2. What was the rational for examining Circ RNA expression? It would have been much better to look at ICP0 or ICP4 expression was because it is known these viral genes are crucial viral regulatory genes.

Reviewer #2: The manuscript is so thorough in describing the construction of the library, but the validation experiment and the single experiment to ascertain the roles of GARP/EARP in viral life cycle are a bit lacking. Figure 7 demonstrates that the genomes/pfu ratio is much higher in cells that lack the interchangeable GARP/EARP complex members, suggesting the model proposed in Figure 8: that trafficking of viral glycoproteins to RE and the TGN is cut off by GARP and EARP KOs, and the vesicles that bud off are deficient in viral glycoproteins essential for 2ndary envelopment of capsids. But Figure 7 - is where the investigation of GARP/EARP begins and ends.

1. Given that these VPS proteins are involved membrane trafficking, it is unclear to what extent virions are produced.

For example, do BHV virions lack gB or other glycoproteins?

BHV viral particles should be purifiable. And also should be GFP+.

Particle:pfu ratios using electron microscopy to show that properly or improperly-formed virions are being made.

Better still, if an Ab exists, immunoelectron microscopy could be performed to show that viral glycoproteins are reduced in the viral envelope.

Are there are normal number of virions in GARP/EARP KO cells? Is their morphology affected?

Could the authors examine membrane protein content of virions from cells deficient in EARP/GARP? Or examine expression of gB and localization of gB at various time points during wt infection of control cells vs. EARP/GARP deficient cells.

2. Lines 363- Strange that HSV-1 results in a relatively tiny (but reproducible) reduction in titer – the fold reduction is not mentioned (and should be) but it looks like a 2-4 fold reduction? This is strangely minimal compared to the 3 log reduction in AIHV titres, a gamma herpesvirus. These seem to be hugely different and at a minimum, worthy of some comment in the discussion. The last sentence says “indicating that AIHV and HSV might employ GARP and EARP similarly to BHV”. If GARP/EARP affect BHV and AIHV titres by 3 logs and HSV titres by 3-fold, are they really functioning similarly?

Line 369 – In fig F, for HSV1, VPS54 -/+ results in the largest reduction in titres at ?? 5X, compared with ? 2.5X reduction with VPS50-52. Yet for AIHV and BHV, the VPS54KO does almost nothing to the AIHV titers (fig H), and the VPS 51-52 reduce titers by 3 logs. This seems to argue that HSV is different from AIHV and BHV. Some discussion is warranted, if not further experimentation.

3. GARP and EARP KO affected viral entry and did not affect viral gene transcription, as ascertained through examination of vDNAout and transcription of IE, E, and L genes.

If transcription of gB is unaffected, and if GARP/EARP affect virion integrity by altering trafficking of gB, then perhaps gB is mislocalized within infected cells?

Reviewer #3: They must, first, assess the expression of viral proteins (in addition to mRNA) to determine the effects of their knock-outs on expression. They must also perform actual virus production assays (i.e., single- and multi-step growth experiments that measure production of infectious virus rather than marker gene expression). Also, there is no TEM or imaging of any kind to support an effect on virus assembly.

**Part III – Minor Issues: Editorial and Data Presentation Modifications**

Reviewer #1: (No Response)

Reviewer #2: 1. The methods describing the expts in Figure 7 are difficult to find and may not exist. In Figure 7BCD they must have infected either clone P or the 52KO cells with BHV1, then taken the supt, pellet, and total, at different time points, and then titer what is collected at these time points on some other type of cell using the methods described in “Plaque assays”. No mention is made of this expt in the methods or legend.

2. Do not understand “n.s.” in Figure D. Error bars are minute or nonexistent? At the 36 hr timepoint, the KO is at 10^3 and the wt is at 1. How could this be n.s., while the differences in C look negligible, with overlapping error bars, and yet are marked with 1 to 4 ****? Not clear to which data points the asterisks in Fig 7C refer. A particular time point only? Which one? Why only one? Why are there error bars only on some points but not others? What is the number of biological replicates?

3. Figure 6 H – cannot find legend for panel H. Must have been intended to be part of panel G. Text references 6H, though.

4. Figure 7 depicts relative genomes, relative mRNA/vDNA, relative vDNA/pfu – relative to what? cannot find or missed description of relativity in legend or Methods.

5. Line 420 – Combined with existing knowledge … - what knowledge is this? References?

6. Line 415 – we conducted a series of experiments to understand the routes these complexes are recruited by the virus for self-benefit. (?? A hybrid of two sentences ?)

7. Line 433-34 referring to colocalization could use some references.

8. Line 336 – would be clearer if text read “VP26-GFP” synthesis (Fig 6D left panel).

9. Line 336 - Figure 6D does not have a left panel…Also the abscissa is unlabeled in Fig 6D top panel –

10. “We also tracked the rate of VP26 synthesis in these KO cells (Fig. 6D left panel) by measuring

GFP fluorescent intensity from 0 h.p.i. to 72 h.p.i, following infection with GFP tagged BHV-1 (Fig. 6D

right panel)” Line 336 - It seems that the difference between the top and bottom panels are the MOI. Both panels depict the KO cell lines, although only the top has a legend. And both panels look like they are following infection with GFP-BHV? This sentence is confusing, as the distinction between left and right panels does not seem to correspond even to the top and bottom panels in Fig 6D.

11. Line 136 and prior – please define btau5.0.1 and rosa26 and TAL1.6.

Reviewer #3: (No Response)

PLOS authors have the option to publish the peer review history of their article (what does this mean?). If published, this will include your full peer review and any attached files.

Reviewer #1: No

Reviewer #2: No

Reviewer #3: No
---

## [Decision Letter · Decision Letter 1]

17 Jul 2023

Dear Dr. Tan,

Thank you very much for submitting your manuscript "GARP and EARP are required for efficient BoHV-1 replication as identified by a genome wide CRISPR knockout screen" for consideration at PLOS Pathogens. As with all papers reviewed by the journal, your manuscript was reviewed by members of the editorial board and by several independent reviewers. The reviewers appreciated the attention to an important topic. Based on the reviews, we are likely to accept this manuscript for publication, providing that you modify the manuscript according to the review recommendations.

Sincerely,

Benjamin E Gewurz, M.D., Ph.D.

Academic Editor

PLOS Pathogens

Patrick Hearing

Section Editor

PLOS Pathogens

Kasturi Haldar

Editor-in-Chief

PLOS Pathogens

orcid.org/0000-0001-5065-158X

Michael Malim

Editor-in-Chief

PLOS Pathogens

orcid.org/0000-0002-7699-2064

Reviewer Comments (if any, and for reference):

Reviewer's Responses to Questions

**Part I - Summary**

Reviewer #2: This manuscript by Tan et al. illustrates the careful construction and impressive documentation of a CRISPR/Cas9 knockout library for high throughput genomics and gene discovery in the cattle genome. The btCRISPR library generated, targeting all protein coding genes in the cattle genome, will be hugely useful in studying cellular processes in bovine cells, as well as processes that intersect with viruses infect bovine cells. At every barrier, the authors went the extra mile to create a high-standard resource. The authors perform extensive gene ontogeny analysis, and illustrate the possible involvement of many pro- and anti-viral host genes involved in the viral replication cycle. They validate one of these sets of genes, comprised of VPS 50-54, which assemble to form the GARP and EARP complexes. They then rigorously knock out the components of these complexes to ascertain their role in the BHV1 life cycle. They arrive at an intriguing model that trafficking of viral glycoproteins to RE and the TGN is cut off by GARP and EARP KOs, and the vesicles that bud off are deficient in viral glycoproteins essential for 2ndary envelopment of capsids, is intriguing and prompts some testable hypotheses. In Figure 8, the authors then test the hypothesis that gB is reduced or absent from released cell-free virions and show that virions generated from VPS52KO possess less gB, supporting the idea that GARP/EARP participate in the normal trafficiking of gB.

Reviewer #3: The revised manuscript of Tan et al., addresses many of the concerns raised in their first submission. The authors have made an effort to address the possibility that there are real effects on virus production due to GARP/EARP subunit knockout (Figure S19) and that that these effects on virus production are due to changes in protein expression rather than effects on assembly (Figures 8 and S20). They are almost there, but there are problems with each of these three critical figures that should be addressed before publication.

Figure S19 is not easily interpretable. The Panel B Y axis is labeled “vDNA copies (relative).” Relative to what? A ratio relative to a cellular housekeeping gene would make sense for total and pellet values, but not for supernatant where the presence of host DNA would indicate cell lysis. The authors should indicate what is actually being measured in this panel. In panel C, the numbers do not make sense. The Y axis is “vDNA/pfu (relative).” If you take any of the time points in panels A and B and calculate the vDNA/pfu ratio, you do not get a value that corresponds to a point in panel C. To give just one example, the clone P 12 h time point for total production, vDNA (relative)/pfu = ~1e2/~1e6 = 1e-4. The value actually plotted in panel C is ~5e0. The trends look about right, but the authors need to explain how the data was actually derived.

The data in Figure S19 is also critical for the conclusions that suthors wish to draw in the paper and should be in the main body of the paper rather than the supplementary data.

Figure S20 does not make sense. First, the western blot in panel A was apparently generated by probing with both anti-gB and anti-GFP simultaneously with no way to distinguish the bands resulting from reactivity with each antibody. For example, there is no way to tell whether the band at about 37k is actually VP26-GFP or a degradation product of the 100k band. Similarly, there is no way to know whether the upper band actually represents reactivity with the putative anti-gB or cross reactivity of the anti-GFP antibody with some other BoHV protein. I cannot understand why the experiment was done in this way. Panels B and C also do not make sense. The box that the authors cut clearly contains the major capsid protein, VP5 (the most abundant protein by copy number in any herpesvirus virion), but they state that gB is the most abundant protein by peptide representation. This is odd to begin with, but their graph appears to show substantially greater abundances for peptides of VP11/12 and 13/14. It is possible that I have mis-read the graph because the authors do not indicate what is on the Y axis (what does “area” mean in this context?). The bottom line is that the figure does not provide evidence that their antibody reacts specifically with gB.

The specificity of this antibody is cast into further doubt by the confocal images shown in Figure 8. They are of too low a resolution to tell for sure (and the lack of a separated red channel does not help), but the putative gB signal appears to be distributed evenly throughout the cytoplasm rather than localized on any membrane structures as would be expected for an envelope glycoprotein.

The bottom line here is that I am unconvinced that this antibody is specifically directed against gB. Demonstration of reactivity in cells that express only gB is necessary to clarify this.

**Part II – Major Issues: Key Experiments Required for Acceptance**

Reviewer #2: (No Response)

Reviewer #3: Demonstration of their putative anti-gB antibody reactivity in cells that express only gB is necessary to show antibody specificity.

**Part III – Minor Issues: Editorial and Data Presentation Modifications**

Reviewer #2: I realize that Figure S20 is the 20th supplementary figure, but I cannot figure it out.

“Western blot using anti BoHV1 Ab and an anti-GFP Ab against concentrated virus samples from TC supt of WT or 51KO cells. Black arrows point to band that matches VP26-GFP.”

This antibody was used to stain lysates of viruses harvested from media “alongside” an anti-GFP antibody.

I don’t understand – is this a GFP blot or a VP26 blot? Or both? If both, I only see one red VP26 band.

Is red anti-GFP or anti-gB? Why would they use both Abs unless you could see them both- w two different colors? Is it supposed to be green as well?

In Fig S20C, there is a color for WT and one for MOCK. But the bars in the graph all look black. I am assuming that the gB, UL47, and gE , which seem to have 2 bars associated with them, the one to the right is gray.

Not clear why MOCK infected cells have viral proteins in them. Is this contamination from neighboring lanes? If so, please explain. Are all those proteins found by MS predicted to be in the 98 kDa range? If not, please discuss. If so, please discuss - just so that the figure is understandable. Confusing. Is the Coomassie stained band at 95 kDa supposed to be gB? Or, was that black box cut out and gB and all the other proteins detected by MS?

Reviewer #3: (No Response)

PLOS authors have the option to publish the peer review history of their article (what does this mean?). If published, this will include your full peer review and any attached files.

Reviewer #2: No

Reviewer #3: No

Figure Files:

Data Requirements:

Reproducibility:

References:

---

## [Editor Report · Decision Letter 2]

13 Nov 2023

Dear Dr. Tan,

We are pleased to inform you that your manuscript 'GARP and EARP are required for efficient BoHV-1 replication as identified by a genome wide CRISPR knockout screen' has been provisionally accepted for publication in PLOS Pathogens.

Best regards,

Benjamin E Gewurz, M.D., Ph.D.

Academic Editor

PLOS Pathogens

Patrick Hearing

Section Editor

PLOS Pathogens

Kasturi Haldar

Editor-in-Chief

PLOS Pathogens

orcid.org/0000-0001-5065-158X

Michael Malim

Editor-in-Chief

PLOS Pathogens

orcid.org/0000-0002-7699-2064
---

## [Editor Report · Acceptance letter]

30 Nov 2023

Dear Dr. Tan,

We are delighted to inform you that your manuscript, "GARP and EARP are required for efficient BoHV-1 replication as identified by a genome wide CRISPR knockout screen," has been formally accepted for publication in PLOS Pathogens.

Best regards,

Michael Malim

Editor-in-Chief

PLOS Pathogens

orcid.org/0000-0002-7699-2064